# Peripheral and central employment of acid-sensing ion channels during early bilaterian evolution

Josep Martí-Solans[1], Aina Børve[2], Paul Bump[3], Andreas Hejnol[2]*[†], Timothy Lynagh[1]*

[1]Michael Sars Centre, University of Bergen, Bergen, Norway; [2]Department of Biological Sciences, University of Bergen, Bergen, Norway; [3]Hopkins Marine Station, Department of Biology, Stanford University, Pacific Grove, United States

**Abstract** Nervous systems are endowed with rapid chemosensation and intercellular signaling by ligand-gated ion channels (LGICs). While a complex, bilaterally symmetrical nervous system is a major innovation of bilaterian animals, the employment of specific LGICs during early bilaterian evolution is poorly understood. We therefore questioned bilaterian animals' employment of acid-sensing ion channels (ASICs), LGICs that mediate fast excitatory responses to decreases in extracellular pH in vertebrate neurons. Our phylogenetic analysis identified an earlier emergence of ASICs from the overarching DEG/ENaC (degenerin/epithelial sodium channel) superfamily than previously thought and suggests that ASICs were a bilaterian innovation. Our broad examination of ASIC gene expression and biophysical function in each major bilaterian lineage of Xenacoelomorpha, Protostomia, and Deuterostomia suggests that the earliest bilaterian ASICs were probably expressed in the periphery, before being incorporated into the brain as it emerged independently in certain deuterostomes and xenacoelomorphs. The loss of certain peripheral cells from Ecdysozoa after they separated from other protostomes likely explains their loss of ASICs, and thus the absence of ASICs from model organisms *Drosophila* and *Caenorhabditis elegans*. Thus, our use of diverse bilaterians in the investigation of LGIC expression and function offers a unique hypothesis on the employment of LGICs in early bilaterian evolution.

*For correspondence:
andreas.hejnol@uib.no (AH);
tim.lynagh@uib.no (TL)

Present address: [†]Friedrich Schiller University Jena, Jena, Germany

Competing interest: The authors declare that no competing interests exist.

## Editor's evaluation

This work examines the evolutionary origins of acid-sensing ion channels (ASICs), a class of pH-sensing receptors expressed throughout the brain and body. By combining analysis of sequences, functional measurements, and measures of tissue distribution, the authors provide solid evidence that ASICs existed far earlier than previously believed. The present data indicate that ASICs emerged after the split between bilaterians (organisms with two-fold symmetry) and Cnidaria (jellyfish, anemones, corals, etc.), approximately 680 million years ago. This evolutionary and functional analysis of ASIC channels across bilaterian lineages provides relevant information about the evolution of nervous and sensory systems.

## Introduction

Morphological and behavioral complexity of animals is facilitated by nervous systems in which peripheral neurons sense and convey information from the environment, and more centralized neurons integrate and dispatch information to effectors (*Arendt, 2021*; *Liebeskind et al., 2016*). Such sensory (environment → cell) and synaptic (cell → cell) signals rely on ligand-gated ion channels (LGICs)

**eLife digest** Most animals on Earth, from worms to chimpanzees, belong to a group known as the bilaterians. Despite their rich variety of shapes and lifestyles, all these creatures share similarities – in particular, a complex nervous system where neurons can quickly relay electric signals. This is made possible by a class of proteins, known as ligand-gated ion channels, which are studded through the membrane of cells. There, they help neurons efficiently communicate with each other by converting external chemical information into internal electrical signals. Yet despite their importance, how and when these proteins have evolved remains poorly understood.

Marti-Solans et al. decided to explore this question by focusing on acid-sensing ion channels, a family which often forms the linchpin of bilaterian neural networks. They examined when these proteins first evolved (that is, in which putative ancestral animals) and where in the body. To do so, they combed through genetic data from all major bilaterian lineages as well as from non-bilaterian groups; this included previously unexplored datasets that give insight into the type of cells in which a particular gene is active.

The analyses revealed that the channels are specific to bilaterians, but that they appeared earlier than previously thought, being present in the very first members of this group. However, at this stage, the proteins were mainly located in cells at the periphery of the body rather than in those from emerging neural circuits. This suggests that the channels were co-opted by nerve cells later on, when the nervous systems became more complex. The proteins being initially located in cells at the outer edge of the body could also explain why they are absent in bilaterian creatures such as fruit flies and nematode worms; these animals all belong to a lineage where growth takes place by shedding their external layers.

Acid-sensing ion channels are an important group of potential drug targets, often being implicated in pain and diseases of the nervous system. The work of Marti-Solans et al. offers an insight into the diversity of roles these proteins can play in the body, demonstrating once again how evolution can repurpose the same biophysical functions to serve a range of needs inside an organism.

(LGICs), membrane proteins that convert chemical messages into transmembrane ionic currents within milliseconds (*Pattison et al., 2019*; *Smart and Paoletti, 2012*). LGICs are found in all multicellular animals (Metazoa) and even in outgroup lineages such as bacteria and plants (*Chen et al., 1999*; *Chiu et al., 1999*; *Tasneem et al., 2005*). But compared to animals without nervous systems (Porifera and Placozoa), animals with nervous systems (Ctenophora, Cnidaria, and Bilateria) present a larger and more diverse LGIC gene content, that is, more Cys-loop receptors, ionotropic glutamate receptors, and degenerin/epithelial sodium channel (DEG/ENaC) genes (*Moroz et al., 2014*). Although the evolution of the original nervous system(s) did not necessarily involve a rapid expansion of LGICs, the elaboration and refinement of nervous systems within particular lineages did, and this likely endowed bilaterian neurons with a sophisticated chemo-electric toolbox in the ancestors of today's complex bilaterian animals (*Arendt, 2021*; *Liebeskind et al., 2015*; *Moroz et al., 2014*). Unfortunately, studies addressing the functional contribution of LGICs to early bilaterians are lacking (*Heger et al., 2020*), and we therefore lack crucial insight into evolution of the nervous system.

Diversity within the DEG/ENaC superfamily of channels exemplifies the novel tools that LGICs can offer an evolving nervous system. Several independent expansions of DEG/ENaC subfamilies have occurred, including degenerin channels (DEGs) in nematodes, pickpocket channels (PPKs) in arthropods, peptide-gated channels (*Hydra vulgaris* Na$^+$ channels [HyNaCs]) in cnidarians, and acid-sensing ion channels (ASICs) in vertebrates (*Assmann et al., 2014*; *Lynagh et al., 2018*; *Matthews et al., 2019*; *Ng et al., 2019*; *Tavernarakis et al., 1997*; *Figure 1A*). ASICs are excitatory sodium channels gated by increased proton concentrations (drops in pH) and are widely expressed in the nervous system of rodents, with scattered expression in other cells, such as epithelia (*Deval and Lingueglia, 2015*; *Figure 1B and C*). ASICs in sensory neurons of skin, joints, and the gastrointestinal tract of rodents and humans contribute to pain, hyperalgesia, and touch (*Holzer, 2015*; *Ikeuchi et al., 2008*; *Jones et al., 2004*; *Price et al., 2001*), typifying a chemosensory role, whereas ASICs expressed postsynaptically in central neurons of rodents mediate depolarization in response to brief drops in synaptic pH during neurotransmission (*Du et al., 2014*; *González-Inchauspe et al., 2017*), indicating

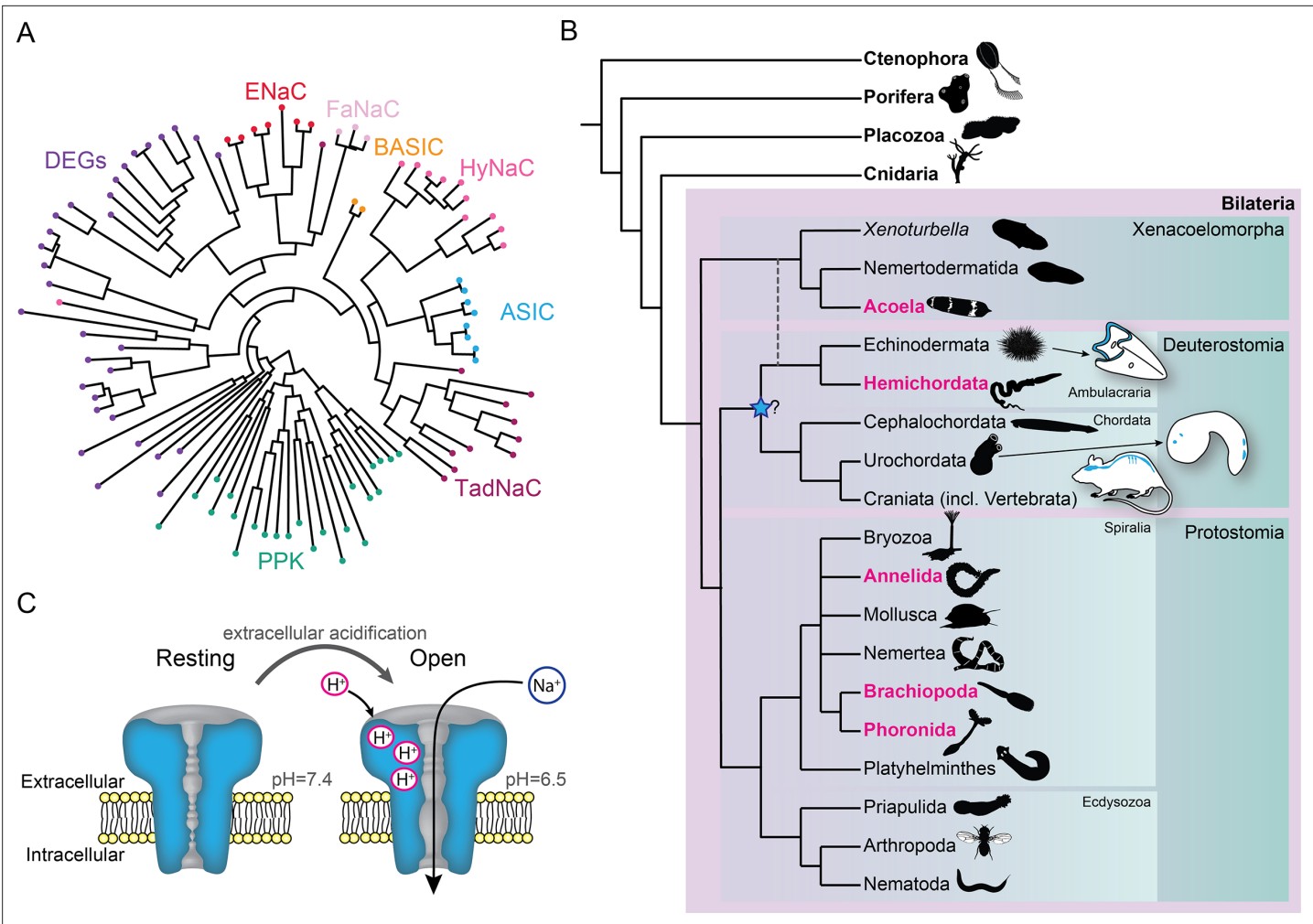

**Figure 1.** Overview of acid-sensing ion channels (ASICs) in Metazoa. (**A**) Abridged phylogenetic tree of the degenerin/epithelial sodium channel (DEG/ENaC) family showing major expansions of previously studied members (after *Assmann et al., 2014*; *Elkhatib et al., 2019*). (**B**) Phylogenetic tree of main animal groups highlighting groups studied in this work (magenta), known ASIC expression (blue), and previously suggested ASIC emergence (blue star). Dotted lines show alternative positions for Xenacoelomorpha. Cartoons show previously described and well-established ASIC protein and/or RNA expression in Chordata (rodent brain, spinal cord, and sensory ganglia, *Foster et al., 2021*; urochordate larva sensory vesicle and bipolar tail neurons, *Coric et al., 2008*) and Ambulacraria (echinoderm larva ciliary band, *Slota et al., 2020*). (**C**) ASIC function. At rest, the channel is closed and impermeable. Upon extracellular acidification, certain amino acid side chains in the ASIC extracellular domain are protonated, causing conformational changes that open the channel, allowing sodium ions to flow down their chemo-electric gradient across the membrane.

an additional, central and synaptic role. ASICs occur in all deuterostomes (*Lynagh et al., 2018*), one of the three major lineages within Bilateria (*Figure 1B*). But given the central or peripheral expression of ASICs in different deuterostomes (*Coric et al., 2008*; *Lynagh et al., 2018*; *Slota et al., 2020*) and uncertainty over the lineage in which ASICs emerged (*Figure 1B*), we lack an accurate assessment of how ASICs were deployed by diversifying bilaterians and thus the chance of better understanding how complex bilaterian nervous systems evolved. This would require a broad and comprehensive assessment of ASICs throughout the major bilaterian lineages.

We therefore performed a thorough phylogenetic investigation of metazoan DEG/ENaC genes, with a focus on ASICs, using unexplored transcriptomes and genomes, revealing ASICs throughout the Bilateria. Moreover, we analyzed gene expression using in situ hybridization and determined the electrophysiological properties of diverse ASICs from each major bilaterian lineage. Results from these experiments enable a new hypothesis on the evolution and function of ASICs during bilaterian evolution.

## Results

### Broader phylogenetic study identifies ASICs in Protostomia and Xenacoelomorpha

We first sought a definitive picture of how broadly ASIC genes are conserved throughout the five metazoan lineages of Bilateria, Cnidaria, Porifera, Placozoa, and Ctenophora (see *Figure 1B* for phylogenetic relationship of these lineages). To this end we utilized previously unexplored transcriptomes combined with canonical resources to search for DEG/ENaC genes from all lineages. The phylogenetic analysis of these 700 sequences from 47 species shows a well-supported clade of ASICs (*Figure 2* and *Figure 2—figure supplement 1*). The ASIC branch consists of two sub-clades 'A' and 'B', both of which include bona fide ASICs from deuterostome bilaterians (*Lynagh et al., 2018*). No ASICs were identified in Cnidaria, Porifera, Placozoa, and Ctenophora.

Bilateria is divided into three major groups, Deuterostomia (above), Protostomia (e.g. arthropods and molluscs), and Xenacoelomorpha (acoelomorph flatworms without nephridia). In contrast to previous studies, we detected ASICs in protostomes and xenacoelomorphs (*Figure 2*). These include putative ASICs from seven protostome species—the annelid *Owenia fusiformis*, the nemertean *Notospermus geniculatus*, the brachiopods *Terebratalia transversa*, *Novocrania anomala* and *Lingula anatina*, the phoronid *Phoronopsis harmeri*, the bryozoan *Membranipora membranacea*, and six xenacolomorph species, the acoels *Hofstenia miamia*, *Isodiametra pulchra*, *Convolutriloba macropyga*, and *Childia submaculatum*, and the xenoturbellans *Xenoturbella bocki* and *Xenoturbella profunda*. Our analysis also included the hemichordate *Schizocardium californicum*, whose ASIC groups with previously reported hemichordate and echinoderm ASICs (*Lynagh et al., 2018*), indicative of broad conservation of ASICs in Ambulacraria, the deuterostome group including hemichordates and echinoderms. This shows that ASIC genes are present in the three major bilaterian groups of deuterostomes, protostomes, and xenacoelomorphs and absent from all other lineages, suggesting that ASICs diverged from other DEG/ENaC genes after the Cnidaria/Bilateria split and before the Bilateria diversified.

Protostomes are divided into Spiralia (such as annelids, molluscs, brachiopods, and phoronids), and Ecdysozoa (such as the nematode *Caenorhabditis elegans* and arthropod *Drosophila melanogaster*) (*Figure 1B*). Notably, the ASIC clade includes no ecdysozoan genes, although our analysis included DEG/ENaC genes from the nematode *Pontonema vulgare*, pan-arthropods *Centruroides sculpturatus* and *Daphnia pulex*, and priapulids *Priapulus caudatus* and *Halicryptus spinulosus*. We also see no putative ASICs in certain spiralians, including the molluscs *Acanthopleura granulata* and *Crassostrea gigas*, the platyhelminths *Prostheceraeus vittatus* and *Schmidtea mediterranea* and in certain xenacoelomorphs, including the nemertodermatids *Meara stichopi* and *Nemertoderma westbladi*. This analysis suggests that ASICs, although found throughout Bilateria, were lost in the lineage to the Ecdysozoa shortly after the Ecdysozoa/Spiralia split and were also lost in certain other bilaterian lineages such as molluscs and nemertodermatids.

Amino acid sequence comparison suggests that diverse ASIC protomers share a conserved structure of a large extracellular domain of ~300 residues between two transmembrane helices, and 30–100 residue intracellular N and C terminal tails. Newly described ASICs are 24–35% identical to rat ASIC1-3, the latter of which share 48–51% identity with each other. The second transmembrane helix is the most conserved segment (38–62%), followed by the extracellular domain (31–39%), with the first transmembrane helix being the most diverse (14–44%). Two clusters of residues in the extracellular domain, one in the acidic pocket and one in the palm/wrist domain, contain protonatable residues that are required for channel activation (*Jasti et al., 2007*; *Paukert et al., 2008*; *Rook et al., 2021*; *Vullo et al., 2017*). Residues H73, D78, E411, and E416 (rat ASIC1a numbering) from the palm/wrist domain and K211 located in an intersubunit palm-thumb interaction region are fairly conserved throughout all ASICs, however conservation of the acidic pocket was only obvious in chordate Group A ASICs, suggesting that this became important for the gating of, for example, rat ASIC1a more recently (*Figure 2—figure supplement 2*).

### ASICs are expressed in two domains in Xenacoelomorpha

The results from the phylogenetic analyses suggest that ASICs emerged during early bilaterian evolution, and we next sought evidence for potential biological roles by investigating the expression of ASIC genes throughout the major bilaterian lineages. The precise relationships between these

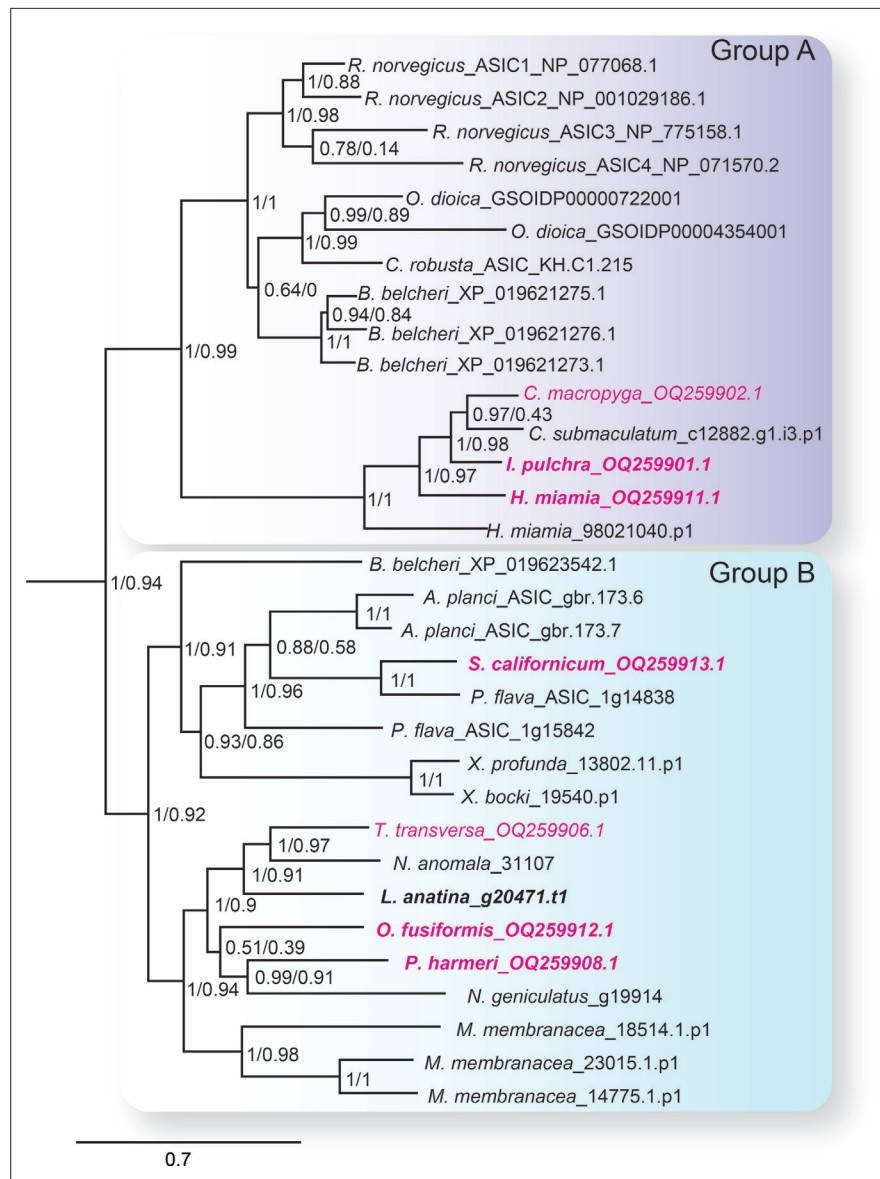

**Figure 2.** The acid-sensing ion channel (ASIC) branch of the degenerin/epithelial sodium channel (DEG/ENaC) family includes only bilaterian genes. ASIC branch from phylogenetic tree of DEG/ENaC family including 700 amino acid sequences from 47 metazoans (*Figure 2—figure supplement 1*). Genes analyzed experimentally in this study are in magenta (gene expression) and/or bold (electrophysiology). Scale bar: amino acid substitutions per site. aBayes (left) and aLRT SH-like (right) likelihood-based support values indicated.

The online version of this article includes the following figure supplement(s) for figure 2:

**Figure supplement 1.** Degenerin/epithelial sodium channel (DEG/ENaC) gene tree.

**Figure supplement 2.** Sequence alignment of relevant domains in acid-sensing ion channels (ASICs).

lineages are under debate, but Xenacoelomorpha forms a putative sister group to all remaining Bilateria (*Cannon et al., 2016*; *Kapli et al., 2021*; *Laumer et al., 2019*; *Philippe et al., 2019*). Therefore, we investigated ASIC expression in Xenacoelomorpha, utilizing the acoels *I. pulchra*, *H. miamia*, and *C. macropyga*. Xenacoelomorphs show a large variation in nervous system architecture, but all possess a basal epidermal nerve plexus and some species, more internally, an anterior condensation of neurons or 'brain' (*Hejnol and Pang, 2016*). *I. pulchra* and *C. macropyga* also possess longitudinal bundles of neurons (*Achatz and Martinez, 2012*; *Martín-Durán et al., 2018*; *Sikes and Bely, 2008*). External to this nervous system in acoels is generally a sheet of longitudinal and ring muscles

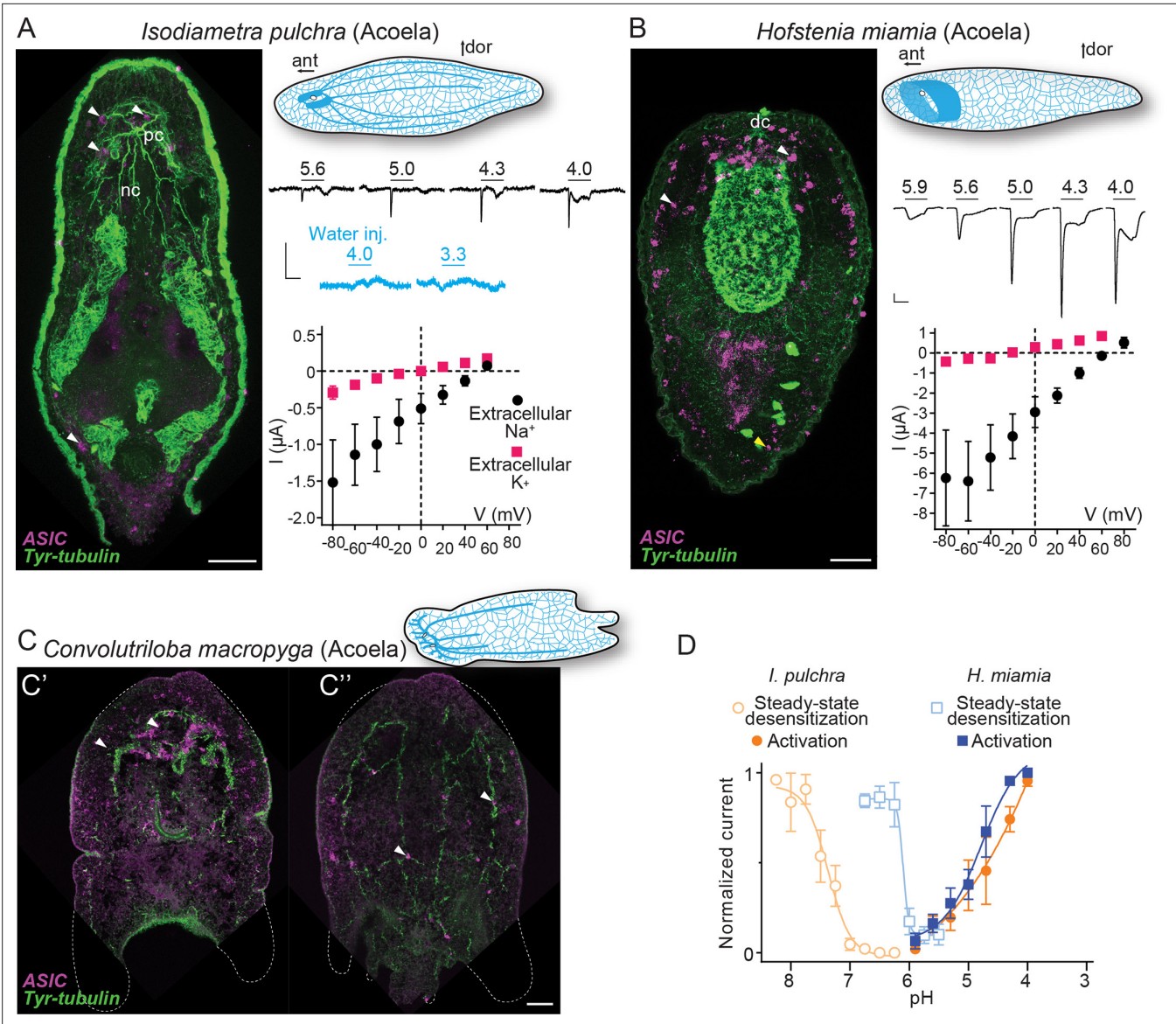

**Figure 3.** Expression and function of acid-sensing ion channels (ASICs) in Xenacoelomorpha. (**A,B**) Left: Fluorescent confocal micrograph showing ASIC mRNA expression (magenta) and tyrosinated tubulin immunoreactivity (green), scale bar: 40 μm. Upper right: Cartoons illustrating morphology and nervous system (blue) after *Martín-Durán et al., 2018*. ant, anterior; dor, dorsal; dc, dorsal commissure; nb, neurite bundles; pc, posterior commissure. Mid-right: Proton-gated currents in xenacoelomorph ASIC-expressing or water-injected *Xenopus laevis* oocytes (scale bars: x, 5 s; y, 0.5 μA). Lower right: Mean (± SEM) pH 4-gated current (I, μA) at different membrane potentials (V, mV) in the presence of 96 mM extracellular NaCl or KCl (n=5–7). Reversal potential ($V_{rev}$) was read off these plots and the difference between $V_{rev,NaCl}$ and $V_{rev,KCl}$ was used to calculate relative ion permeability ($P_{Na+}/P_{K+}$, *Materials and methods*). (**C**) ASIC mRNA expression (magenta) and tyrosinated tubulin immunoreactivity (green) in *Convolutriloba macropyga*. As *C. macropyga* is larger, images of slightly ventral (**C'**) and dorsal (**C''**) planes are shown. (**D**) Filled symbols: Mean (± SEM) normalized current amplitude in response to increasing proton concentrations (activation, n=6–8). Open symbols: Mean (± SEM) normalized current amplitude in response to pH 4 following pre-incubation in decreasing pH (steady-state desensitization, n=5–6).

The online version of this article includes the following source data and figure supplement(s) for figure 3:

**Source data 1.** Numerical data contributing to *Figure 3*.

**Figure supplement 1.** Expression of neuronal markers in Xenacoelomorpha: *Isodiametra pulchra* (**A**), *Hofstenia miamia* (**B**), and *Convolutriloba macropyga* (**C**).

and, more externally, ciliated epithelial cells mediating locomotion (*Haszprunar, 2016*; *Martin, 1978*; *Raikova et al., 2016*).

From a dorsal view, expression of *I. pulchra* ASIC mRNA can be detected in several cells across the central anterior of the adult animal (*Figure 3A*). This is the location of the *I. pulchra* brain, where, for example, serotonergic and peptidergic neurons form lateral lobes, connected by a frontal ring and a posterior commissure, and from which four pairs of nerve cords extend posteriorly (*Figure 3A*; *Achatz and Martinez, 2012*). Fluorescent confocal micrographs show that the ASIC-expressing cells are associated with the posterior commissure of the brain and the dorsal and lateral neurite bundles, including very posterior positions (*Figure 3A*, white arrowheads). Double fluorescent in situ hybridization of ASIC with *choline acetyltransferase* (*ChAT*) probes revealed that part of ASIC-expressing cells overlap with few cholinergic neurons of the brain (*Figure 3—figure supplement 1A*). In *H. miamia*, we observed high ASIC expression in cells throughout the anterior third of the animal (*Figure 3B*), correlating with the brain-like anterior condensation of neurons in this acoel (*Hulett et al., 2020*). Double fluorescent in situ hybridization showed ASIC localization in two anterior cell populations: GABAergic neurons expressing glutamate decarboxylase and directly adjacent cells (*Figure 3—figure supplement 1B*). This might reflect emerging single-cell RNA sequencing data on *H. miamia* identifying three distinct ASIC-expressing cell types: two muscle-like and one neuron-like (*Hulett et al., 2022*). Similar to *I. pulchra*, the ASIC signal is strongest near the dorsal commissure but, in contrast, also in scattered cells across the animal, including in the very periphery (*Figure 3B*, yellow arrowhead). *C. macropyga* ASIC expression was high and broad. In the anterior, ASIC-expressing cells are clearly associated with the brain (*Figure 3C'*, white arrowheads) and potentially in the same cells expressing the neuronal marker *sine-oculis like* 3/6 (*Six3/6*) (*Figure 3—figure supplement 1C*). Throughout the animal, including the very periphery, there is medium to high ASIC expression, perhaps associated with locomotory ciliated cells covering the animal or the extensive nerve plexus (*Figure 3C''*). In summary, all Xenacoelomorpha species investigated here exhibit expression in two domains, in the brain, as well as dispersed, peripheral expression. Thus, ASICs may have had central, integrative, and peripheral sensory functions in early acoels.

## Xenacoelomorph ASICs mediate excitatory currents gated by high proton concentrations

When ASICs are exposed to drops in extracellular pH they typically show a transient depolarizing current (inward flow of $Na^+$ ions), rapidly followed by either desensitization—a non-ion-conducting state in the presence of agonist—or a smaller sustained current (*Krishtal, 2015*). To test the function of xenacoelomorph ASICs, we injected the ASIC cRNAs into *Xenopus laevis* oocytes and measured membrane current in response to decreasing pH using two-electrode voltage clamp. At oocytes expressing *I. pulchra* ASIC, drops to pH 5.6 and lower rapidly activated a transient current (*Figure 3A*, mid-right). However, compared to other ASICs, responses to increased proton concentrations were relatively inconsistent at *I. pulchra* ASIC: although transient currents were never activated by pH higher than 5.9, the concentration dependence of the transient current between pH 5.6 and 4.0 was inconsistent and at lower pH was usually followed by a sustained current ~25% the amplitude of the transient current (*Table 1*). To verify that these variable currents were indeed specific to the heterologous *I. pulchra* channels, we also applied pH 4.0 to oocytes from the same batch injected with water (*Figure 3A*, mid-right) or with a proton-insensitive channel (HyNaC2/4/6; *Figure 2—figure supplement 1*), and we did not observe pH 4.0-gated currents (*Figure 2—figure supplement 1*). Thus, *I. pulchra* ASIC is indeed gated by protons, but with relatively low potency (pH for half-maximal activation of transient current ($pH_{50}$)=4.9 ± 0.2, *Table 1*). The *H. miamia* ASIC OQ259911.1 also formed homomeric proton-gated channels, with rapid transient currents followed by smaller sustained currents (18% the amplitude of transient current) in response to drops to pH 5.6 through 4.0 ($pH_{50}$=4.8 ± 0.1, *Figure 3B and D*, *Table 1*). The presence of another ASIC gene in this species, 9021040 in *Figure 2*, raises the possibility of heteromeric channels in Xenacoelomorpha, but we have not pursued that here. Finally, we observed no proton-gated currents in *Xenopus* oocytes injected with *C. macropyga* ASIC cRNA (n=6), suggesting either low heterologous expression or proton insensitivity of this channel.

Like vertebrate ASICs, *I. pulchra* and *H. miamia* ASICs showed decreased responses to activating pH (4.0) after pre-incubation in slightly decreased pH (*Figure 3D*, open symbols, *Table 1*), indicating that steady-state desensitization is a broadly conserved phenomenon. Similarly, xenacoelomorph

**Table 1.** Biophysical characteristics of bilaterian acid-sensing ion channels (ASICs) (mean ± SEM).

| Animal | Proton sensitivity | | | | | | | | | | | | | Permeability | |
|---|---|---|---|---|---|---|---|---|---|---|---|---|---|---|---|
| | $I_{max}$ (Low-$Ca^{2+}$, µA) | n | $pH50_a$ (Low-$Ca^{2+}$) | n | $I_{max}$ (µA) | n | $pH50_a$ | n | $pH50_{SSD}$ | n | $T_{50\%}(s)$* | n | $I_{sus}/I_{tra}$† | n | $P_{Na+}/P_{K+}$ | n |
| **Xenacoelomorph** | | | | | | | | | | | | | | | | |
| *Isodiametra pulchra* | 3.6±0.8 | 5 | 5.3±0.2 | 7 | 0.8±0.2 | 5 | 4.9±0.2 | 8 | 7.4±0.1 | 5 | 0.09±0.02 | 6 | 0.25±0.04 | 6 | 8.3±2.0 / ≥8†§ | 7 / 6‡ |
| *Hofstenia miamia* | 46.1±11.6 | 5 | 5.6±0.1 | 6 | 7.3±1.5 | 5 | 4.8±0.1 | 4 | 6.1±0.1 | 6 | 0.18±0.02 | 6 | 0.18±0.03 | 6 | 30.1±5.9 / 27.7±6.2† | 5 / 4‡ |
| **Spiralian** | | | | | | | | | | | | | | | | |
| *Lingula anatina* | 32.1±4.8 | 6 | 7.5±0.01 | 5 | 11.5±1.9 | 6 | 7.3±0.2 | 6 | 7.7±0.03 | 4 | 8.96±0.42 | 5 | 0.02±0.01 | 5 | 9.0±1.7 | 8 |
| *Phoronopsis harmeri* | 10.4±0.6 | 5 | 5.7±0.3 | 10 | 7.0±1.3 | 5 | 5.2±0.1 | 10 | 6.5±0.05 | 5 | 0.22±0.02 | 4 | 0.03±0.01 | 4 | 3.0±1.0 | 5 |
| *Owenia fusiformis* | 56.6±11.1 | 6 | 8.3±0.1 | 7 | 21.3±2.7 | 6 | 8.1±0.1 | 5 | 8.6±0.02 | 6 | 0.15±0.01 | 5 | 0.00±0.00 | 4 | 12.0±0.3 | 6 |
| **Hemichordate** | | | | | | | | | | | | | | | | |
| *Schizocardium californicum* | 25.8±3.3 | 6 | 6.05±0.3 | 3 | 7.9±2.0 | 6 | 5.3±0.1 | 6 | 7.4±0.17 | 3 | 0.18±0.02 | 4 | 0.01±0.00 | 6 | 2.9±0.5 | 5 |

*Time to half-maximal desensitization (half the time from peak transient current to plateaued, sustained current).

†Relation between current amplitudes for sustained and transient peaks ($I_{sus}/I_{tra}$).

‡Values for the sustained current.

§For *I. pulchra* ASIC, sustained current with extracellular NaCl clearly reversed at 60 mV, and sustained currents with extracellular KCl were small, generally outward, and difficult to quantify. Together this leads us to conclude that $P_{Na+}/P_{K+}$ is high.

The online version of this article includes the following source data for table 1:

**Source data 1.** Numerical data contributing to *Table 1* that is not already in Figure Source Data files.

ASICs share the property of inhibition by extracellular calcium with vertebrate ASICs. *I. pulchra* and *H. miamia* ASICs showed 0.4 and 0.8 unit increases in $pH_{50}$ (increased potency) and four- to sixfold increases in current amplitude when the extracellular calcium concentration was reduced from 1.8 to 0.1 mM (*Table 1*).

Most ASICs so far studied have a slight preference for sodium over potassium ions, with a relative sodium/potassium ion permeability ($P_{Na+}/P_{K+}$) of ~10, and thus activation of ASICs leads to depolarization and generation of action potentials in mouse neurons (*Gruol et al., 1980*). *I. pulchra* ASIC ($P_{Na+}/P_{K+}=8.3 \pm 2.0$) showed similar $Na^+$ selectivity to most ASICs (*Figure 3A*, lower right and *Table 1*), whereas *H. miamia* $Na^+$ selectivity ($P_{Na+}/P_{K+}=30 \pm 5.9$; *Figure 3B*) is remarkably high compared to most ASICs and is more reminiscent of the high preference for sodium seen in ENaCs (*Gründer and Pusch, 2015*). For both acoel ASICs, the relative ion permeability was similar for transient and sustained currents (*Table 1*). Since pore-lining and other amino acid residues implicated in ion selectivity in vertebrate ASICs are conserved in xenacoelomorph ASICs, the higher $Na^+$ preference of *H. miamia* ASIC must derive from yet unidentified determinants (*Figure 2—figure supplement 2*).We conclude, however, that both in xenacoelomorphs, which belong to the putative sister group to all other bilaterians, and in vertebrates such as rodents and humans, ASICs are excitatory receptors for protons that are expressed highly in both the brain and in the periphery.

## ASICs are expressed in peripheral neurons and the digestive system in Spiralia

Excitatory proton-gated currents and combined central and peripheral expression of xenacoelomorph ASICs is reminiscent of vertebrate ASICs. However, whether the ancestral bilaterian ASIC performed similar roles remains unclear, as the position of Xenacoelomorpha as sister taxon of all other bilaterians or as part of a group with Ambulacraria within Deuterostomia is debated (*Cannon et al., 2016*; *Kapli et al., 2021*; *Laumer et al., 2019*; *Philippe et al., 2019*; *Figure 1B*). We therefore turned to the third major lineage of bilaterians and investigated previously unidentified ASICs in Protostomia. We investigated the expression and function of ASICs in brachiopod, phoronid, and annelid larvae, allowing us to study whole-animal gene expression in animals with typical spiralian features such as an anterior brain or apical organ and ventrolateral or peripherally extending nerve cords.

In the brachiopod *T. transversa*, ASIC was expressed most highly in cells at the lateral edge of the central neuropil in late larvae (*Figure 4A*). These cells are slightly lateral/caudal to previously identified sensory neurons in the *T. transversa* apical organ (*Santagata et al., 2012*), and additional in situ hybridization targeting distinct neuronal types showed that the ASIC-expressing cells are close to or overlap with cholinergic neurons at the lateral edge of the central neuropil (*Figure 4— figure supplement 1A*). Confocal images suggest that these ASIC-expressing cells project latero-anteriorly (*Figure 4A*, lateral view), perhaps indicative of an anterior sensory role. In the phoronid *P. harmeri*, ASIC was expressed in two principal domains. There was a clear signal colocalizing with *synaptotagmin*-expressing cells in the perimeter of the hood (*Figure 4B*, white arrowheads), a domain innervated by the peripheral nerve ring and controlling swimming by ciliary beating (*Marinković et al., 2020*; *Temereva and Tsitrin, 2014a*). In stark contrast to *synaptotagmin*, ASIC was not expressed in the more central parts of the nervous system, including both the apical organ, an anterior group of sensory neurons that gives way to the developing brain, and the main nerve ring that runs caudally from the apical organ through the trunk and innervates the tentacles (*Figure 4— figure supplement 1B*; *Nielsen, 2015*; *Temereva and Tsitrin, 2014a*; *Temereva and Tsitrin, 2014b*). The second domain, in which expression was even more prominent, was the digestive system, where ASIC colocalized with the endodermal marker *Gata*. Here, the ASIC expression was high in numerous cells around the digestive tract (comprising a stomach in the trunk and intestine in the pedicle at this stage), most noticeably just below the pharynx and around the lower half of the stomach (*Figure 4B*, orange arrowheads and inset, *Andrikou et al., 2019a*). We also detected digestive system expression in the more distantly related spiralian, the annelid *O. fusiformis*. In 19 days post fertilization *O. fusiformis*, we observed no ASIC expression in the young nervous system consisting of a dorsal apical organ already expressing the neuronal marker *six3/6* (*Figure 4C*), ventral prototroch (or ciliary band) neurons, and anterior and lateral connecting neurons (*Carrillo-Baltodano et al., 2021*; *Helm et al., 2016*). Instead, ASIC was clearly expressed in cells around the oesophagus (*Figure 4C*). In the lumen of the midgut some extracellular signal appeared repeatedly (*Figure 4C*). These results show that

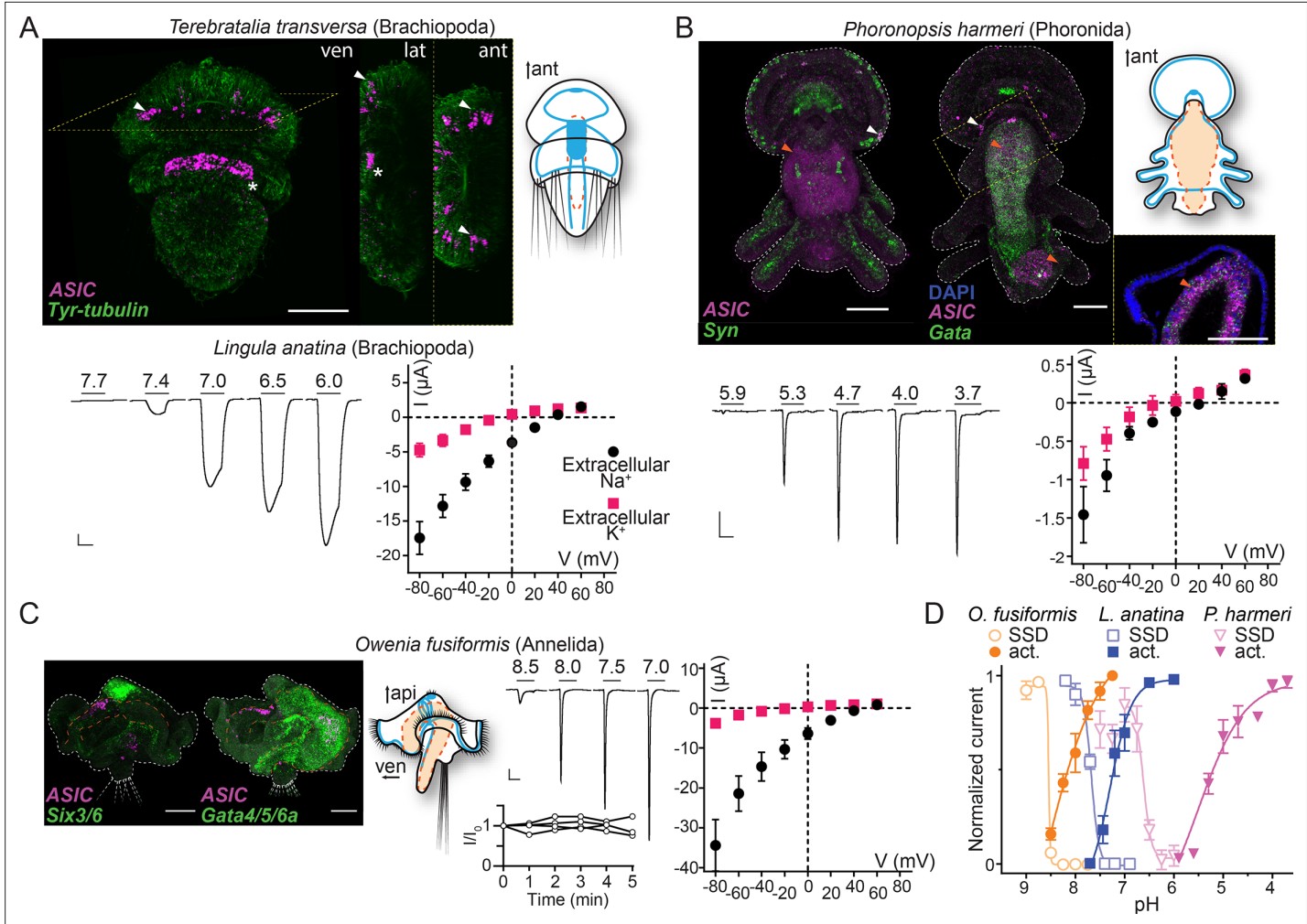

**Figure 4.** Expression and function of acid-sensing ion channels (ASICs) in Spiralia. (**A,B**) Upper left: ASIC mRNA expression (magenta) and the neuronal (*synaptotagmin*, *Syn*; *sine-oculis like 3/6*, *Six3/6*; in green) and endodermal markers (*Gata*, in green). Tyrosinated tubulin immunoreactivity (green) (ant, anterior; api, apical; lat, lateral; ven, ventral; scale bar: 40 μm). Arrowheads, ASIC expression; asterisks, unspecific staining common in *Terebratalia transversa* (**Martín-Durán et al., 2018**). Yellow dashed frame images are magnifications of the yellow dashed line boxes. Upper right: Cartoons illustrating nervous system (blue) and digestive system (orange; Anlage only in *T. transversa*) after **Gąsiorowski et al., 2021**; **Helm et al., 2016**. Lower left: Proton-gated currents in *Xenopus laevis* oocytes expressing indicated spiralian ASICs (scale bars: x, 5 s; y, 1 μA). Lower right: Mean (± SEM) pH 6.5- (*Lingula anatina*) or 4- (*Phoronopsis harmeri* ASIC) gated current (I, μA) at different membrane potentials (V, mV) in the presence of 96 mM extracellular NaCl or KCl (n=5–8). Reversal potential ($V_{rev}$) was read off these plots and the difference between $V_{rev,NaCl}$ and $V_{rev,KCl}$ was used to calculate relative ion permeability ($P_{Na+}/P_{K+}$). (**C**) Left to right: ASIC expression, animal morphology, proton-gated currents (top), and normalized current amplitude, that is, amplitude of the current at certain time (-I) divided by the current amplitude at the start of the experiment ($I_0$), in response to same proton concentration (pH 7.0) in oocytes perfused with pH 9.0 solution for 5 min (bottom), lines connect data from individual oocytes, and pH 6.5-gated current at different membrane potentials, as in (**A,B**) at *Owenia fusiformis* ASIC. (**D**) Filled symbols: Mean (± SEM) normalized current amplitude in response to increasing proton concentrations (activation, 'act.', n=6–10). Open symbols: Mean (± SEM) normalized current amplitude in response to pH 7 for *O. fusiformis*, 6.5 for *L. anatina*, and 4 for *P. harmeri* ASIC following pre-incubation in decreasing pH (steady-state desensitization, 'SSD', n=4–5).

The online version of this article includes the following source data and figure supplement(s) for figure 4:

**Source data 1.** Numerical data contributing to *Figure 4*.

**Figure supplement 1.** Expression of neuronal and endodermal markers in Spiralia: *Terebratalia transversa* (**A**), *Phoronopsis harmeri* (**B**), and *Owenia fusiformis* (**C**).

in various spiralians, ASICs are found in the periphery and the digestive system, not in the brain or apical organ.

## Spiralian ASICs show a wide range of proton sensitivity and ion selectivity

We next tested the electrophysiological properties of spiralian ASICs by expressing them heterologously in *Xenopus* oocytes. We observed no proton-activated currents in *Xenopus* oocytes injected with *T. transversa* (brachiopod) ASIC RNA (n=10), and we cannot conclude if this is due to low heterologous expression or unknown function of this channel. However, we identified an ASIC transcript in another brachiopod, *L. anatina* (*Luo et al., 2015*), and observed functional expression of this channel. *L. anatina* ASIC was sensitive to relatively low proton concentrations, with large currents in response to pH 7.4 and lower that enter desensitization slower than most ASICs (*Figure 4A*, $pH_{50}$=7.3 ± 0.2, time to 50% current amplitude ($T_{50\%}$)=8.96 ± 0.42 s, *Table 1*) and showed typical $Na^+/K^+$ selectivity (*Figure 4A*, $P_{Na+}/P_{K+}$=9.0 ± 1.7). *L. anatina* ASIC has a cysteine residue in contrast to an asparagine residue at position 414 (rat ASIC1a numbering) of vertebrate ASICs (brown in *Figure 2—figure supplement 2A*). This cysteine residue might contribute to the slow desensitization of *L. anatina* ASIC, as the N414C mutation in human ASIC1a (rat ASIC1a numbering) slows desensitization slightly and its chemical modification slows desensitization greatly (*Roy et al., 2013*).

The phoronid *P. harmeri* ASIC showed much lower proton sensitivity, activated by pH in the range of 5.9–3.7, with rapidly desensitizing currents (*Figure 4B*, $pH_{50}$=5.2 ± 0.1, *Table 1*) and almost no preference for sodium over potassium ions (*Figure 4B*, $P_{Na+}/P_{K+}$=3.0 ± 0.9, *Table 1*). We next tested the function of ASIC from the annelid *O. fusiformis*. This channel was extremely sensitive to low proton concentrations. When held at pH 7.5 and exposed to lower pH, *O. fusiformis* ASIC showed very small current responses, but when held at pH 9.0 and exposed to small drops in pH, large inward currents were activated that rapidly desensitized (*Figure 4C*, $pH_{50}$=8.1 ± 0.1, *Table 1*). We measured repeated responses of *O. fusiformis* ASIC to pH 7.0 over the course of 5 min in pH 9.0 and observed no decrease in current amplitude (*Figure 4C*), verifying that the relatively basic resting pH of 9.0 was not harming oocytes and skewing results with this channel. *O. fusiformis* ASIC also showed ion selectivity typical of most ASICs (*Figure 4C*, $P_{Na+}/P_{K+}$=12.0 ± 0.3). We also found that reducing the extracellular calcium concentration from 1.8 to 0.1 mM increased the $pH_{50}$ and increased current amplitude at *O. fusiformis* ASIC (*Table 1*), indicating the extreme proton sensitivity of this channel is not due to an absence of inhibition by extracellular calcium. *P. harmeri* and *L. anatina* ASICs were also inhibited by extracellular calcium, with 0.2 to 0.5 unit increases in $pH_{50}$ and two- to threefold increases in current amplitude in 0.1 mM extracellular calcium relative to 1.8 mM (*Table 1*).

Spiralian ASICs seem to vary in apparent proton affinity and in ion selectivity among the lineages, from low proton sensitivity (pH ~5) and essentially non-selective cation currents in phoronid ASIC to high proton sensitivity (pH 7–8) and ~10-fold selective sodium permeability in brachiopod and annelid ASICs. All showed canonical steady-state desensitization, occurring in pH ranges higher than activating pH (*Figure 4D* and *Table 1*). Spiralian ASIC genes thus encode proton-gated cation channels, and ASICs are present in all major groups of bilaterians: Xenacoelomorpha, Protostomia, and Deuterostomia.

## Hemichordate ASIC is expressed in peripheral cells and pharynx and mediates rapidly desensitizing excitatory currents in response to protons

Regarding Deuterostomia, combined expression and function of ASICs have so far only been characterized in selected chordates (urochordates and vertebrates) (*Coric et al., 2008*; *Kellenberger et al., 2015*; *Lynagh et al., 2018*), and relatively little is known about ASICs in the ambulacrarian lineage (echinoderms and hemichordates). Recent studies on neural development in sea urchin larvae (*Lytechinus variegatus*, an echinoderm) showed that ASIC is expressed diffusely throughout the ciliary band (*Slota et al., 2020*). This transcript (MH996684) is closely related to the Group B ASICs from echinoderms and hemichordates that showed proton-gated currents when expressed heterologously (*Lynagh et al., 2018*). Here, we used the hemichordate *S. californicum* to characterize the combined expression and function of an ambulacrarian ASIC. *S. californicum* ASIC expression was visible in late larval stages when structures such as the nervous system and the ciliary bands become more

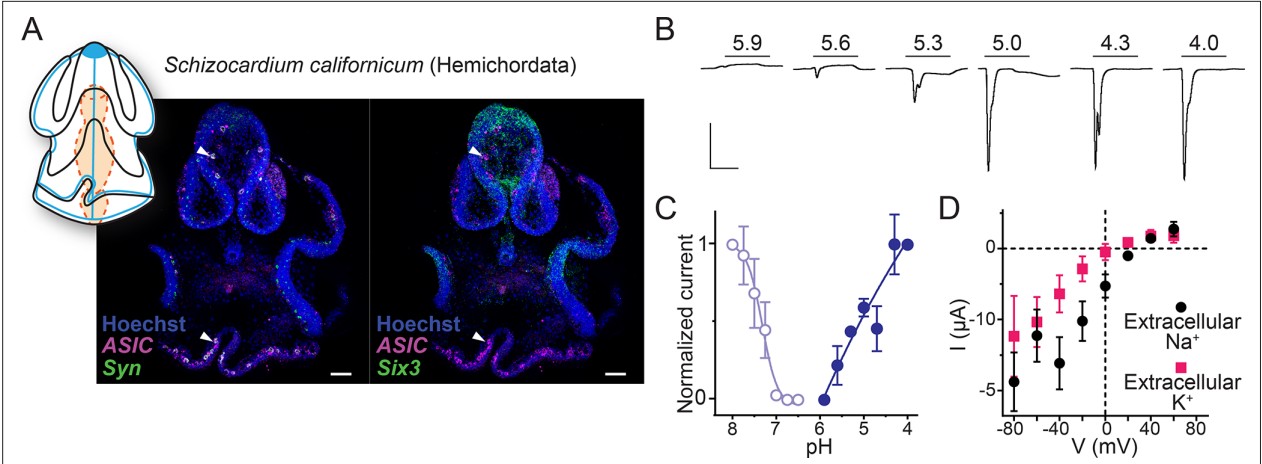

**Figure 5.** Expression and function of hemichordate acid-sensing ion channel (ASIC). (**A**) Cartoon illustrating nervous system (blue) and digestive system (orange) in *Schizocardium californicum*, after *Gonzalez et al., 2017*, and *hybridization chain reaction* (HCR) showing expression of *S. californicum* ASIC and the neuronal markers *synaptotagmin* (Syn) or *sine-oculis like* (*Six3*). White arrowheads highlight ASIC expression. Colocalization of two genes' expression is visually identified in the regions of the image that appear white. Nuclei are stained blue with Hoescht. Scale bar: 50 μm. (**B**) Proton-gated currents in *Xenopus laevis* oocytes expressing *S. californicum* ASIC. Scale bars: x, 5 s; y, 2 μA. (**C**) Filled symbols: Mean (± SEM) normalized current amplitude in response to increasing proton concentrations (activation, 'act.', n=3). Open symbols: Mean (± SEM) normalized current amplitude in response to pH 4 following pre-incubation in decreasing pH (steady-state desensitization, n=4). (**D**) Mean (± SEM) pH 4-gated current (I, μA) at different membrane potentials (V, mV) in the presence of 96 mM extracellular NaCl or KCl (n=5).

The online version of this article includes the following source data and figure supplement(s) for figure 5:

**Source data 1.** Numerical data contributing to *Figure 5*.

**Figure supplement 1.** Expression of acid-sensing ion channel (ASIC) and neuronal markers in Hemichordata: Colorimetric in situ hybridization for ASIC (**A**) and *hybridization chain reactions* (HCRs) for *ASIC* (**B**), *synaptotagmin* (Syn) (**C**), *sine-oculis like* (*Six3*) (**D**), and the merge (**E**).

intricate, like sea urchin ASIC (*Slota et al., 2020*). The tornaria larva of *S. californicum* has two main ciliary bands: the circumoral band for feeding, and the telotroch, innervated by serotonergic neurites, for locomotion (*Gonzalez et al., 2017*). We performed numerous ASIC in situs in *S. californicum* larvae and observed variable expression patterns between the colorimetric and fluorescent in situ (*Figure 5—figure supplement 1*). Expression in the ciliary band was consistent but there was also signal in other tissues such as the pharynx that varied across samples (*Figure 5—figure supplement 1A*). To resolve these inconsistencies, we turned to HCR (*hybridization chain reaction*, *Choi et al., 2018*) to better resolve the expression pattern. By doing this we were able to resolve the variability of the ASIC in situ across experiments as well as specificity of the expression pattern by including the markers *synapotagmin* which marks neural cells (*Nakajima et al., 2004*) and *Six3* which marks the most anterior territory including the apical organ in *S. californicum* (*Gonzalez et al., 2017*). With these triple HCRs, we found ASIC expressed in some of the *synaptotagmin* cells (*Figure 5A*, white arrowhead, and *Figure 5—figure supplement 1B–E*) but excluded from the most anterior, apical organ, *Six3*-positive territory.

When heterologously expressed, *S. californicum* ASIC showed rapidly activating and desensitizing currents in response to extracellular acidification with proton sensitivity comparable to a previously described hemichordate ASIC and lower than echinoderm ASICs (*Figure 5B*, pH$_{50}$=5.3 ± 0.1) (*Lynagh et al., 2018*). The *S. californicum* channel showed relatively weak ion selectivity, with P$_{Na+}$/P$_{K+}$ values slightly above unity (2.9±0.5; *Figure 5D* and *Table 1*), similar to ion selectivity in previously described ambulacrarian ASICs (*Lynagh et al., 2018*). Finally, current amplitude and proton potency were both inhibited by extracellular calcium, with 0.7 unit increased pH$_{50}$ and fourfold larger currents in 0.1 mM calcium compared to 1.8 mM calcium (*Table 1*). *S. californicum* thus presents a functional ASIC that is expressed in the peripheral nervous system and the digestive system, and it appears that such peripheral expression of ASICs is a conserved feature of Deuterostomia, Protostomia, and Xenacoelomorpha.

## Discussion

### The emergence of ASICs

Our results show that ASICs—of the same family as the prototypical rat ASIC1a—are present in Xena-coelomorpha and Protostomia in addition to Deuterostomia and are thus conserved in the three major groups of Bilateria. We also find, consistent with previous studies, that ASICs of this family are absent from the other major lineages of Cnidaria, Placozoa, Porifera, and Ctenophora, indicating that ASICs emerged in the lineage to the Bilateria, soon after the Cnidaria/Bilateria split ~680 Mya (*Kumar et al., 2017*). The clades most closely related to ASICs within the DEG/ENaC superfamily tree include three from which several channels have been characterized: mammalian bile acid-sensitive ion channels (BASICs), *Trichoplax adhaerans* Na⁺ channels (TadNaCs); and HyNaC and *Nematostella vectensis* Na⁺ channels (NeNaCs). We cannot establish phylogenetically the identity of the ancestral gene from which ASICs emerged, as branch support toward the base of the ASIC + BASIC + TadNaC + HyNaC/NeNaC clade is relatively low: our maximum likelihood (ML) trees inferred with aLRT SH-like and aBayes statistics yielded slightly different topologies within this clade (*Figure 2—figure supplement 1C and D*), and previous studies using similar methods to each other also inferred slightly different topologies within this clade (*Aguilar-Camacho et al., 2022*; *Elkhatib et al., 2022*). Statistical support for the distinct ASIC clade is strong, however (*Figure 2—figure supplement 1*).

We are thus left to consider the *functional* relationships among these cousins. BASICs, highly expressed in mammal intestines, are activated by bile acids or are constitutively active, and show inhibition by protons (*Wiemuth et al., 2014*; *Wiemuth and Gründer, 2010*). Non-mammalian genes from the BASIC clade have not been characterized. HyNaCs are neuropeptide-gated channels from the medusozoan cnidarian *Hydra* (*Assmann et al., 2014*), and when NeNaCs from the anthozoan cnidarian *Nematostella* were recently reported, surprisingly, NeNaC2 and NeNaC14 were activated by protons (pH₅₀ values of 5.8 and <4.0), not by cnidarian neuropeptides (*Aguilar-Camacho et al., 2022*). The TadNaC clade also includes a proton-activated channel, TadNaC2 (pH₅₀ 5.1), in addition to a proton-inhibited channel, TadNaC6 (*Elkhatib et al., 2019*; *Elkhatib et al., 2022*). Thus, proton-activated and -inhibited channels occur sporadically throughout the ASIC +

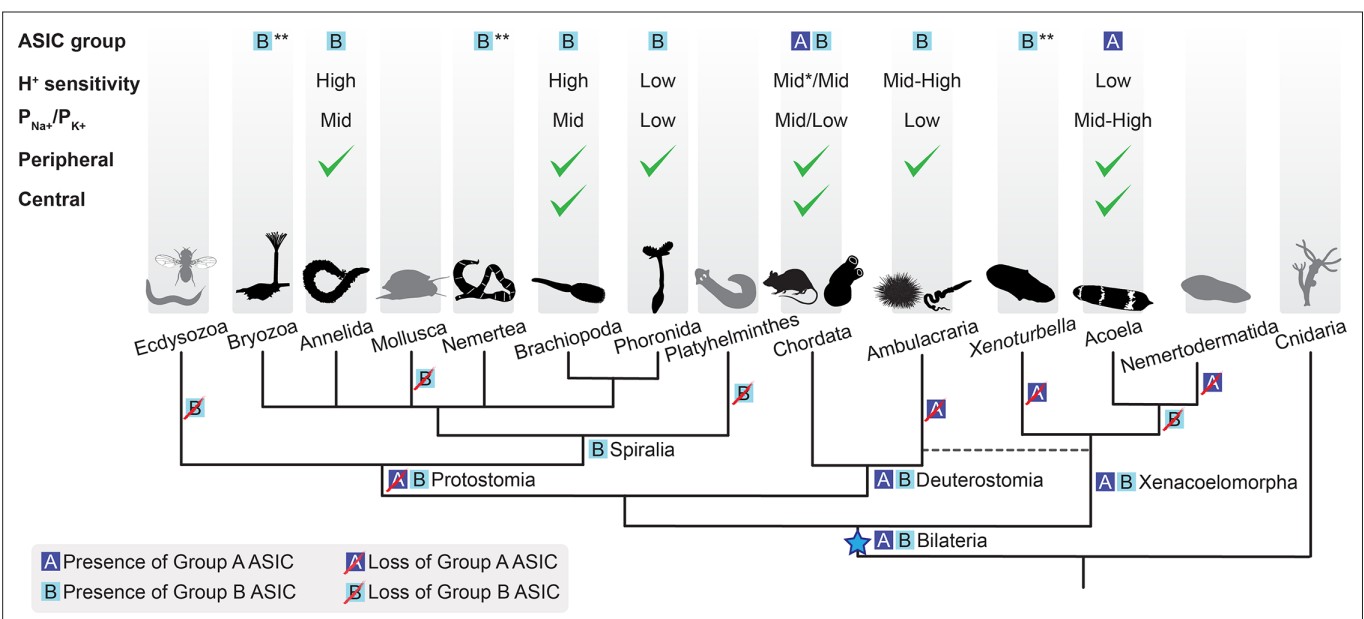

**Figure 6.** Evolutionary history of acid-sensing ion channel (ASIC) function in Metazoa. Upper half of diagram indicates characteristic properties of ASIC in different lineages: degenerin/epithelial sodium channel (DEG/ENaC) gene tree position ('ASIC group A or B'); proton (H⁺) sensitivity; relative ion permeability ('P_Na+/P_K+'); native expression pattern ('Peripheral', ciliary or gastrointestinal epithelia and/or peripheral neurons; 'Central', brain and/or nerve cords). Lower half shows phylogenetic relationships of the different animal phyla studied here. Blue star, putative emergence of ASICs in the last common ancestor of all bilaterians after Cnidaria/Bilateria split. For H⁺ sensitivity, 'low', 'mid', and 'high' correspond to pH₅₀ <5.3, 5.3–7, and >7, respectively. *, a minority of characterized chordate ASICs have low H⁺ sensitivity (*Gründer and Pusch, 2015*; *Paukert et al., 2004*). For relative ion permeability, 'low', 'mid', and 'high' correspond to P_Na+/P_K+≤3, 3–15, and >15, respectively. **, channel function not experimentally tested.

BASIC + TadNaC + HyNaC/NeNaC clade, and only in the ASIC clade is proton-induced activation the defining feature. ASICs could thus have emerged from a proton-activated ancestor, after which this function was selected for; or from a peptide- or bile acid-activated ancestor, in which a novel mechanism of activation evolved. Certain functional and phylogenetic data have led others to favor the latter possibility (*Elkhatib et al., 2022*; *Golubovic et al., 2007*), and indeed structural features important for channel gating throughout the prototypical ASIC family, such as highly conserved histidine residues and/or aromatic interactions, are absent from proton-gated NeNaCs and TadNaCs (*Elkhatib et al., 2022*; *Lynagh et al., 2018*). The fact that proton-induced activation has emerged in numerous DEG/ENaC channels, even outside the ASIC + BASIC + TadNaC + HyNaC/NeNaC clade, as shown for a *Drosophila* PPK channel and three *C. elegans* degenerin-like channels (*Jang et al., 2019*; *Kaulich et al., 2022*), suggests a propensity for this function in the DEG/ENaC superfamily.

Soon after its emergence in an early bilaterian, we infer that ASIC duplicated, giving rise to two similar proton-gated channels, based on Group A and Group B ASICs in our phylogeny. Subsequently, and at different times, most descendants lost one of these (*Figure 6*). Acoels lost Group B ASIC and xenoturbellans lost Group A ASIC soon after this early split within the Xenacoelomorpha. In contrast, the first deuterostomes and, even more recently, the first chordates likely retained both ASICs, reflected in the continued presence of both in Cephalochordata (*Figure 2*, *Figure 6*). Subsequently, in Olfactores (Craniata + Urochordata), Group B ASIC was lost and Group A ASIC underwent independent radiations, leading to, for example, ASIC1-ASIC4 paralogues in vertebrates (*Paukert et al., 2004*), two Group A ASICs in urochordates (*Lynagh et al., 2018*), and multiple Group A ASICs in cephalochordates (*Figure 2*).

## Peripheral and central roles for ASICs

By investigating ASIC gene expression in each of the three major bilaterian lineages, we see that ASICs are principally found: centrally, in the brain; and/or peripherally, in the digestive system, peripheral nerve rings, and/or scattered peripheral cells. This, together with previous research on ASICs in rodents, points to at least two functions of ASICs: a central signaling role, where ASICs mediate rapid excitatory signals between neurons in the brain (*Du et al., 2014*) and a peripheral sensory/modulatory role, where ASICs convert increased proton concentrations in the environment—extra-organismal or in the gut—into excitatory neuronal signals (*Krishtal and Pidoplichko, 1981*). Brain expression of mouse, zebrafish, and urochordate ASICs suggests conservation of the synaptic role in various chordates (*Du et al., 2014*; *González-Inchauspe et al., 2017*; *Lynagh et al., 2018*; *Paukert et al., 2004*), whereas in the other deuterostome lineage of Ambulacraria (*Figure 1B*), nervous systems are less centralized and ASIC expression appears more peripheral (*Slota et al., 2020* and *Figure 6*). In Xenacoelomorpha, based on the markers we used, we could not assign ASIC expression to a particular type of neuron, except in *I. pulchra* and *H. miamia*, where some of the ASIC-expressing cells were cholinergic and GABAergic, respectively (*Figure 3—figure supplement 1A and B*). In contrast, central ASICs were observed in only one of the three Spiralia tested, and even these ASIC-expressing cells showed projections toward the very anterior of the animal, perhaps indicative of a less integrative and more sensory role (*Figure 4A*). Brain ASICs are thus common to Xenacoelomorpha and Deuterostomia and are less prevalent in Protostomia, and this central role is more commonly played by Group A ASICs than Group B ASICs (*Figure 6*).

The second, peripheral domain, observed in the seven species of Xenacoelomorpha, Spiralia, and Deuterostomia studied here, largely corresponds to the ciliation in the species. While xenacoelomorphs are completely covered with ciliated epithelial cells, in many other Bilateria the ciliation is restricted to ciliary bands or the ciliary ventral region, both used for locomotion. These ASICs are thus expressed in a location to sense external changes in pH and directly modulate ciliary movement via excitatory current or, via depolarization and synaptic release onto more central neurons, send the information centrally. Conversely, if these ASICs are expressed on the proximal side of these peripheral cells, their role may be modulation of ciliary function in response to efferent modulatory signals. Although activation of postsynaptic ASICs has so far been linked to only glutamatergic and GABAergic synapses (*Du et al., 2014*; *Storozhuk et al., 2016*) and innervation of ciliary cells appears monoaminergic in Spiralia (*Marinković et al., 2020*), monoaminergic vesicles are also acidic and could thus foreseeably release protons to activate ASICs here (*Onoa et al., 2010*).

We were surprised to see expression of ASIC predominantly in the digestive system of the spiralians *O. fusiformis* and *P. harmeri*. In mammals, chemosensors such as ASIC and transient receptor potential channels (e.g. TRPV1) are expressed in sensory neurons innervating the gastrointestinal tract (*Holzer, 2015*), but oesophageal, intestinal, and lung epithelial cells also express ASICs, potentially contributing to acid-induced secretions, transport, and inflammation (*Dong et al., 2011*; *Su et al., 2006*; *Ustaoglu et al., 2021*). The gastrointestinal expression of ASICs that we observed may reflect a chemosensory role, like that mediated by ASICs in neurons innervating the lungs and skin of mammals (*Diochot et al., 2012*; *Gu and Lee, 2006*). Similar functions are tentatively suggested by ASIC expression in the peripheral nervous system of Ambulacraria (*Slota et al., 2020*; *Figure 5*). Again, we can't exclude the possibility that these peripheral ASICs are modulating ciliary function, either directly via acid-induced activation or indirectly, via efferent signals from central neurons. Nonetheless, these expression patterns seem to suggest that ASICs play a sensory role in the periphery of Deuterostomia, Protostomia, and Xenacoelomorpha via expression in cells that correlate with ciliation, indicating that such a role was probably present in the ancestral bilaterian. Consistent with the early appearance of this role, it is played by both Group A and Group B ASICs, which both emerged in early bilaterians. Subsequently, ASICs—particularly Group A—were likely deployed to the other, more central role for inter-neuronal communication in lineages such as Acoela and Chordata.

## Loss of ASICs

The conservation and high expression of ASICs in bilaterians begs the question as to why they were lost in certain lineages. The reason for the loss of ASICs in selected Spiralia, such as molluscs and platyhelminthes is unclear, especially when one considers that numerous molluscs are susceptible to the tide and thus vast fluctuations in pH. Presumably, gene radiations that increased sensory ion channel or transporter diversity in such animals have compensated for the loss of ASICs (*Albertin et al., 2015*; *Fu et al., 2021*; *Simakov et al., 2013*; *Xun et al., 2020*; *Zhang et al., 2012*), and indeed, certain members of various channel families are capable of mediating excitation (or inhibition) in response to decreased pH (*Pattison et al., 2019*). Perhaps most noticeable, however, is the absence of ASICs in Ecdysozoa, a broad lineage including pan-arthropods, nematodes, and priapulids, each of which we considered in our phylogenetic analysis. Ecdysozoa split from Spiralia ~650 Mya (*Kumar et al., 2017*), adopting a chitinous cuticle utilizing rigid locomotory structures and requiring periodic molting for growth (*Howard et al., 2020*; *Schmidt-Rhaesa et al., 1998*). Concomitantly, Ecdysozoa lost the ectodermal, motile ciliated cells inherited from the last common ancestor of Cnidaria and Bilateria that mediate locomotory, feeding, secretory, and sensory functions in most other bilaterians (*Ringers et al., 2020*; *Valentine and Collins, 2000*). The correlation between loss of ASICs and loss of motile ciliated cells, together with the conservation of other ciliated and sensory cells in Ecdysozoa, indicates that the role of ASICs in early Protostomia was primarily associated with ectodermal ciliated cells rather than more central sensory cells associated with, for example, mouth, eyes, and antennae. This, together with the presence of ASICs in the periphery of Spiralia, Xenacoelomorpha, and Deuterostomia, suggests that the role of the earliest ASIC was likely local conversion of external chemical stimuli into modulation of locomotion, conversion of central signals into modulation of locomotion, or both. Indeed, motile ciliated epithelia in human lungs are modulated by both external and central stimuli (*Shah et al., 2009*), although the precise contribution of ASICs in these cells is unclear (*Su et al., 2006*).

## Outlook

We acknowledge that future experiments might benefit from greater genomic resources for non-bilaterian animals, but our comprehensive survey of DEG/ENaC genes from Ctenophora, Porifera, Placozoa, and Cnidaria finds no ASICs in those lineages. Our study suggests that ASICs emerged in an early bilaterian, most likely in peripheral cells or epithelia, and were gradually adopted into the central nervous system of certain complex animals. This offers a unique insight into the employment of LGICs during early bilaterian evolution. Functional characterization of diverse ASICs shows a considerable breadth of pH sensitivity and ion selectivity throughout the family, offering new tools for probing the biophysical mechanisms of function. The combined use of gene expression and experimental analysis is thus a useful tool in understanding protein evolution and function.

# Materials and methods

**Key resources table**

| Reagent type (species) or resource | Designation | Source or reference | Identifiers | Additional information |
|---|---|---|---|---|
| gene (*Isodiametra pulchra*) | ASIC | NCBI | OQ259901.1 | |
| gene (*Isodiametra pulchra*) | ChAT | NCBI | KY709762.1 | |
| gene (*Isodiametra pulchra*) | VAchT | NCBI | KY709763.1 | |
| gene (*Isodiametra pulchra*) | VMAT | NCBI | KY709764.1 | |
| gene (*Isodiametra pulchra*) | Syn | NCBI | OQ259900.1 | |
| gene (*Isodiametra pulchra*) | TH | NCBI | KY709765.1 | |
| gene (*Isodiametra pulchra*) | TpH | NCBI | KY709766.1 | |
| gene (*Hofstenia miamia*) | ASIC | NCBI | OQ259911.1 | |
| gene (*Hofstenia miamia*) | Gad-1 | NCBI | MT657938.1 | |
| gene (*Hofstenia miamia*) | TpH-1 | NCBI | MT657942.1 | |
| gene (*Convolutriloba macropyga*) | ASIC | NCBI | OQ259902.1 | |
| gene (*Convolutriloba macropyga*) | Six3/6 | NCBI | OQ259903.1 | |
| gene (*Convolutriloba macropyga*) | TH | NCBI | OQ259904.1 | |
| gene (*Convolutriloba macropyga*) | PH | NCBI | OQ259905.1 | |
| gene (*Terebratalia transversa*) | ASIC | NCBI | OQ259906.1 | |
| gene (*Terebratalia transversa*) | ChAT | NCBI | KY809754.1 | |
| gene (*Terebratalia transversa*) | VAchT | NCBI | KY809753.1 | |
| gene (*Terebratalia transversa*) | TH | NCBI | OQ259907.1 | |
| gene (*Terebratalia transversa*) | TpH | NCBI | KY809752.1 | |
| gene (*Lingula anatina*) | ASIC | OIST | g20471.1 | |
| gene (*Phoronopsis harmeri*) | ASIC | NCBI | OQ259908.1 | |
| gene (*Phoronopsis harmeri*) | Six3/6 | NCBI | MN431430.1 | |
| gene (*Phoronopsis harmeri*) | TH | NCBI | OQ259909.1 | |
| gene (*Phoronopsis harmeri*) | Syn | NCBI | OQ259910.1 | |
| gene (*Phoronopsis harmeri*) | Gata4/5/6 | NCBI | MN431425.1 | |
| gene (*Owenia fusiformis*) | ASIC | NCBI | OQ259912.1 | |
| gene (*Owenia fusiformis*) | Six3/6 | NCBI | KR232531 | |
| gene (*Owenia fusiformis*) | Gata4/5/6 a | NCBI | KR232537 | |
| gene (*Schizocardium californicum*) | ASIC | NCBI | OQ259913.1 | |
| gene (*Schizocardium californicum*) | Syn | NCBI | OQ259914.1 | |

*Continued on next page*

*Continued*

| Reagent type (species) or resource | Designation | Source or reference | Identifiers | Additional information |
|---|---|---|---|---|
| gene (*Schizocardium californicum*) | Six3 | NCBI | KX845335.1 | |
| commercial assay or kit | pGEM-T Easy vector | Promega | A1360 | |
| commercial assay or kit | Ambion Megascript T7 | Invitrogen | AM1334 | |
| commercial assay or kit | Ambion Megascript SP6 | Invitrogen | AM1330 | |
| commercial assay or kit | TSA Cy3 kit | PerkinElmer | NEL744001KT | |
| commercial assay or kit | TSA Cy5 kit | PerkinElmer | NEL745001KT | |
| antibody | Monoclonal Anti-Tubulin, Tyrosine antibody produced in mouse | Sigma-Aldrich | T9028 | |
| antibody | Alexa Fluor 488 goat anti-mouse IgG | Life Technologies | A-11029 | |
| antibody | Anti-Digoxigenin-AP, Fab fragments | Roche | 11093274910 | |
| antibody | Anti-Digoxigenin-POD, Fab fragments | Roche | 11207733910 | |
| antibody | Anti-DNP HRP Conjugate | Akoya | TS-0004000 | |
| other | DAPI | Molecular Probes | D1306 | |
| sequence-based reagent | Scal_ASIC | Molecular Instruments, Inc | HCR probe | |
| sequence-based reagent | Scal_Six3 | Molecular Instruments, Inc | HCR probe | |
| sequence-based reagent | Scal_Syn | Integrated DNA Technologies | DNA oligo pool | |
| other | HCR amplifiers with fluorophores B1-Alexa Fluor-647 | Molecular Instruments, Inc | | |
| other | HCR amplifiers with fluorophores B2-Alexa Fluor-488 | Molecular Instruments, Inc | | |
| other | HCR amplifiers with fluorophores B3-Alexa Fluor-546 | Molecular Instruments, Inc | | |
| recombinant DNA reagent | modified pSP64poly(A) (plasmid) | This paper | | contains 5'- and 3'-UTR sequences of the *X. laevis* β-globin gene, and a C-terminal Myc tag |
| commercial assay or kit | SP6 Polymerase mMessage mMachine kit, Ambion | Fisher | 10391175 | |
| biological sample (*Xenopus laevis*) | Oocytes | Ecocyte Bioscience | | |
| other | OC-725C amplifier | Warner Instruments | | |
| other | LIH 8+8 digitizer | HEKA | | |
| software | Patchmaster | HEKA | | |
| software | pClamp v10.7 | Molecular Devices | | |
| chemical compound, drug | Sodium ursodeoxycholic acid | Santa Cruz Biotechnology | sc-222407 | |
| peptide, recombinant protein | (pyroE)WLGGRFamide | Genscript | | |

## Survey and phylogenetic analysis

Mouse ASIC1a was used as a query in tBLASTn searches of DEG/ENaC genes in xenacoelomorphs (*C. macropyga, C. submaculatum, H. miamia, I. pulchra, X. bocki, X. profunda, N. westbladi*, and *M. stichopi* from transcriptomes published in **Andrikou et al., 2019b**), spiralians (*Spadella* spp., *Dimorphilus gyrociliatus, Epiphanes senta, Lepidodermella squamata, Lineus longissimus, Lineus ruber, M.*

*membranacea*, *N. anomala*, *O. fusiformis*, *P. harmeri*, *P. vittatus*, and *T. transversa* from our transcriptomes in preparation; *A. granulata*, *C. gigas*, and *Brachionus plicatilis* from NCBI; *L. anatina* and *N. geniculatus* from OIST; and *S. mediterranea* from SmedGD), ecdysozoans (*H. spinulosus*, *P. caudatus*, *P. vulgare* from our transcriptomes; *D. melanogaster*, *D. pulex*, *C. sculpturatus*, and *C. elegans* from NCBI) and a hemichordate (*S. californicum*, our transcriptome). Other DEG/ENaC genes were retrieved via BlastP at public databases NCBI, Compagen, JGI, OIST, OikoBase, Aniseed, or UniProt targeting cnidarians (hexacorallian *N. vectensis*, octacorallian *Dendronephthya gigantea*, scyphozoan *Aurelia aurita*, hydrozoan *H. vulgaris*), poriferan (*Amphimedon queenslandica*), the placazoan *Trichoplax adhaerens*, ctenophores (cydippid *Pleurobrachia bachei* and lobate *Mnemiopsis leidyi*), and deuterostomes (chordates *Rattus norvegicus*, *Ciona robusta*, *Oikopleura dioica*, and *Branchiostoma belcheri*; hemichordate *Ptychodera flava*; and echinoderm *Acanthaster planci*). Amino acid sequences were aligned using MAFFT (**Katoh and Standley, 2013**), variable N- and C-termini were removed and highly similar sequences were not considered (**Supplementary file 1**), and homologies were assigned by phylogenetic tree analyses based on ML inferences calculated with PhyML v3.0 (**Guindon et al., 2010**). Robustness of tree topologies was assessed under automatic model selection based on Akaike information criteria. Due to computational load of bootstrap performance, trees were inferred using the fast likelihood-based methods: aLRT SH-like; and aBayes (**Anisimova et al., 2011**). Cell marker genes (**Supplementary file 2**) were identified similarly, but whereas homology of synaptotagmin, Gata, and Six3 was obvious, homology of PH, TH, and TpH genes was confirmed phylogenetically with reference to earlier work (**Siltberg-Liberles et al., 2008**).

## Animal collection and fixation

Stable cultures of acoels were maintained in the laboratory. *C. macropyga* (**Shannon and Achatz, 2007**) were reared in a tropical aquarium system with salinity 34±1 ppt at a constant temperature of 25°C. The aquariums were illuminated (Pacific LED lamp WT470C LED64S/840 PSU WB L1600, Philips) on a day/night cycle of 12/12 hr. The worms were fed with freshly hatched brine shrimp *Artemia* twice per week. *I. pulchra* (**Smith and Bush, 1991**) were cultured as described by **De Mulder et al., 2009**, and *H. miamia* (**Correa, 1960**) as described by **Srivastava et al., 2014**. For the remaining species, adult gravid animals were collected from Bodega Bay, California, USA (*P. harmeri* **Pixell, 1912**), San Juan Island, Washington, USA (*T. transversa* **Sowerby, 1846**), Station Biologique de Roscoff, France (*O. fusiformis* **Delle Chiaje, 1841**), and Kanangra Boyd National Park and Morro Bay State Park, California, USA (*S. californicum* **Cameron and Perez, 2012**). Animals were spawned and larvae obtained as described in **Freeman, 1993**; **Gonzalez et al., 2018**; **Rattenbury, 1954**; **Wilson, 1932**. Adult and larval specimens were starved for 2–7 days prior to fixation. Samples were relaxed in 7.4% magnesium chloride and fixed in 4% paraformaldehyde in culture medium for 1 hr at room temperature and washed several times in 0.1% Tween 20 phosphate buffered saline (PBS), dehydrated through a graded series of methanol, and stored in pure methanol or ethanol at −20°C.

## Cloning

The full-length coding sequences of identified ASIC genes were amplified from cDNA of *C. macropyga*, *I. pulchra*, *H. miamia*, *P. harmeri*, *O. fusiformis*, *T. transversa*, and *S. californicum* by PCR using gene-specific primers. PCR products were purified and cloned into a pGEM-T Easy vector (Promega, A1360) according to the manufacturer's instructions and the identity of inserts confirmed by sequencing. Riboprobes were synthesized with Ambion Megascript T7 (AM1334) and SP6 (AM1330) kit following the manufacturer's instruction for subsequent in situ hybridization. Additional cell markers (**Supplementary file 2**) were cloned similarly, whether full-length or shorter, as desired for in situ hybridization probes (below).

## Immunohistochemistry and situ hybridization

Single whole-mount colorimetric and fluorescent in situ hybridization was performed following an established protocol (**Martindale et al., 2004**) with probe concentration of 0.1 ng/µl (*I. pulchra*) or 1 ng/µl (the remaining species) and hybridization temperature of 67°C. Proteinase K treatment time was adjusted for each species and ranged from 2 min (*P. harmeri*, *O. fusiformis*, *S. californicum*) to 10 min (*T. transversa*). Post-hybridization low salt washes were performed with 0.05× saline sodium citrate (SSC; *H. miamia*) or 0.2× SSC (the remaining species). Fluorescent in situ hybridization was

visualized with TSA Cy3 kit (PerkinElmer, NEL752001KT). Samples were mounted in 70% glycerol or subjected to immunohistochemistry for visualization of neural structures: samples were permeabilized in 0.2% Triton X in PBS (PTx) and blocked in 1% bovine serum albumin in PTx (PBT) and incubated with antibodies against tyrosinated tubulin (Sigma, T9028) at a concentration of 1:250 in PTx with 5% normal goat serum, and incubated for 16–18 hr at 4°C. After several washes in PBT the samples were incubated with secondary goat anti-mouse antibodies conjugated with Alexa Fluor 488 (Life Technologies), at a concentration 1:200 in PTx with 5% normal goat serum for 16–18 hr at 4°C, and samples washed extensively before mounting in 70% glycerol and imaging. Nuclei were stained with DAPI (Molecular Probes), unless otherwise indicated. When results were consistent (most animals), in situ hybridization was performed at least twice. For ambiguous results (*S. californicum*), several more rounds were performed, and the most representative results were chosen for display in figures.

## Hybridization chain reaction

HCR in situ hybridization was performed in *Schizocardium* following an established protocol (*Bump et al., 2022*). For HCR probe design, complementary DNA sequences for *ASIC* and *Six3* were submitted to Molecular Instruments, Inc and for *synaptotagmin*, the Ozpolat Lab HCR probe generator (*Kuehn et al., 2022*) was used and sequences were ordered as a DNA oligo pool from Integrated DNA Technologies. HCR amplifiers with fluorophores B1-Alexa Fluor-647, B2-Alexa Fluor-488, and B3-Alexa Fluor-546 were ordered from Molecular Instruments, Inc.

## Imaging

Representative specimens from colorimetric in situ hybridization experiments were imaged with a Zeiss Axiocam 503 color connected to a Zeiss Axioscope 5 using bright-field Nomarski optics. Fluorescently labeled samples were scanned on an Olympus FV3000 confocal laser-scanning microscope and Leica SP8 confocal laser-scanning microscope. Colorimetric in situ stained with antibodies were scanned in a Leica SP5 confocal laser-scanning microscope using reflection microscopy protocol as described by *Jékely and Arendt, 2007*. Images were analyzed with Imaris 9.8.0 and Photoshop CS6 (Adobe), and figure plates were assembled with Illustrator CC. Brightness/contrast and color balance adjustments were applied to the whole image, not parts.

## Electrophysiological recordings and data analysis

For expression in *X. laevis* oocytes and electrophysiological experiments, coding sequences were mutated synonymously to remove internal restriction sites if necessary and subcloned into SalI and XbaI sites of a modified pSP64poly(A) vector (ProMega), containing 5′ SP6 sequence, 5′- and 3′-UTR sequences of the *X. laevis* β-globin gene, and a C-terminal Myc tag, with an EcoRI restriction site after the poly(A) tail (*Supplementary file 3*). *L. anatina* ASIC (g20471.t1 from *L. anatina* Ver 2.0, OIST Marine Genomics Unit), synonymously mutated to remove internal restriction sites and including a C-terminal myc tag before the stop codon, was commercially synthesized and subcloned (Genscript) into HindIII and BamHI sites of pSP64poly(A) (*Supplementary file 3*). Plasmids were linearized with EcoRI, and cRNA was synthesized in vitro with SP6 Polymerase (mMessage mMachine kit, Ambion). Stage V/VI *X. laevis* oocytes, purchased from Ecocyte Bioscience (Dortmund, Germany), were injected with 3–90 ng cRNA. After injection, oocytes were incubated for 1–3 days at 19°C in 50% Leibowitz medium (Merck) supplemented with 0.25 mg/ml gentamicin, 1 mM L-glutamine, and 15 mM HEPES (pH 7.6). Whole-cell currents were recorded from oocytes by two-electrode voltage clamp using an OC-725C amplifier (Warner Instruments) and an LIH 8+8 digitizer with Patchmaster software (HEKA), acquired at 1 kHz and filtered at 200 Hz. Currents were also analyzed in pClamp v10.7 software (Molecular Devices) and additionally filtered at 10 Hz (*I. pulchra* ASIC, which has faster kinetics) or 1 Hz (for all other ASICs) for display in figures. Oocytes were clamped at –60 mV, unless otherwise indicated, and continuously perfused with a bath solution containing (in mM): 96 NaCl, 2 KCl, 1.8 $CaCl_2$, 1 $MgCl_2$, and 5 HEPES (for pH >6.0) or 5 MES (for pH ≤6.0). pH was adjusted with NaOH, HCl, or KOH, as appropriate. Low-$Ca^{2+}$ bath solution contained (in mM): 96 NaCl, 2 KCl, 0.1 $CaCl_2$, and 5 HEPES. In most experiments, activating/desensitizing pH was applied to oocytes in-between resting periods (at pH 7.5 for most ASICS, at pH 9.0, 8.6, and 8.0 for *O. fusiformis*, *L. anatina*, *S. californicum* ASICs, respectively, unless otherwise indicated) of at least 30 s. After retrieving current amplitude from pClamp, all data analyses were performed in Prism v9 (GraphPad Software). In concentration-response graphs, currents

are normalized to maximum proton-gated current. For ion selectivity experiments, IV relationships were measured in regular bath solution and that in which extracellular NaCl was replaced with KCl. IV relationships were obtained by activating the channels at different membrane potentials from –80 to 60 mV, with 20 mV increments, unless otherwise indicated. Reversal potentials ($V_{rev,Na+}$ and $V_{rev,K+}$) were taken from the intersection of the IV curve with the voltage axis. These values were used to calculate relative permeability $P_{Na+}/P_{K+}$ with the Goldman-Hodgkin-Katz equation, $P_{Na+}/P_{K+}=\exp(F(V_{rev,Na+} - V_{rev,K+})/RT)$, where F=Faraday constant, R=gas constant, and T=293 K. Standard chemicals were purchased from Merck. Specialist chemicals (*Figure 2—figure supplement 1*) were purchased from Santa Cruz Biotechnology (item sc-222407, sodium ursodeoxycholic acid, ≥98% purity) or synthesized by Genscript ((pyroE)WLGGRFamide, ≥97% purity—'Hydra RFamide I' from *Assmann et al., 2014*).

## Acknowledgements

We thank Chris Lowe (Stanford University) for *P harmeri* and *S californicum* samples, Nadia Rimskaya-Korsakova, Kevin Pang, Chema Martín-Durán, and Carmen Andrikou for sharing cell marker clones. This work was supported by The Research Council of Norway, project number 234817.

## Additional information

### Funding

| Funder | Grant reference number | Author |
| --- | --- | --- |
| Research Council of Norway | 234817 | Andreas Hejnol<br>Timothy Lynagh |

The funders had no role in study design, data collection and interpretation, or the decision to submit the work for publication.

### Author contributions

Josep Martí-Solans, Conceptualization, Formal analysis, Funding acquisition, Investigation, Visualization, Methodology, Writing – original draft, Writing – review and editing; Aina Børve, Resources, Formal analysis, Investigation, Visualization, Methodology, Writing – review and editing; Paul Bump, Resources, Formal analysis, Investigation, Visualization, Methodology, Writing – original draft, Writing – review and editing; Andreas Hejnol, Formal analysis, Supervision, Investigation, Visualization, Methodology, Writing – original draft, Writing – review and editing; Timothy Lynagh, Conceptualization, Formal analysis, Supervision, Funding acquisition, Methodology, Writing – original draft, Writing – review and editing

### Author ORCIDs

Josep Martí-Solans http://orcid.org/0000-0002-7016-5789
Andreas Hejnol http://orcid.org/0000-0003-2196-8507
Timothy Lynagh http://orcid.org/0000-0003-4888-4098

### Ethics

This study used simple invertebrate animals that do not require University of Bergen ethics approval.

### Decision letter and Author response

Decision letter https://doi.org/10.7554/eLife.81613.sa1
Author response https://doi.org/10.7554/eLife.81613.sa2

## Additional files

### Supplementary files

• Supplementary file 1. Amino acid sequence alignment used for acid-sensing ion channel (ASIC) phylogenies.

• Supplementary file 2. DNA sequences of neuronal and endodermal genes utilized in in situ hybridization or *hybridization chain reaction* gene expression studies. Aromatic amino acid

hydroxylase gene tree.

• Supplementary file 3. Modified pSP64 vector and inserts used for *Xenopus laevis* expression and electrophysiology.

• MDAR checklist

### Data availability

The amino acid sequence alignment used to generate the phylogenetic tree is available in Supplementary file 1. All DNA sequences used for in situ hybridization and hybridization chain reaction are available in Supplementary file 2 or Supplementary file 3. All DNA sequences used for electrophysiological experiments are available in Supplementary file 3. Figure 3—source data 1, Figure 4—source data 1, and Figure 5—source data 1 contain the numerical data used to generate the respective figures. Table 1—source data 1 contains much of the numerical data used to generate Table 1 (the rest is already included in the other source data files).

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
