## [Editor Report]

This work examines the evolutionary origins of acid-sensing ion channels (ASICs), a class of pH-sensing receptors expressed throughout the brain and body. By combining analysis of sequences, functional measurements, and measures of tissue distribution, the authors provide solid evidence that ASICs existed far earlier than previously believed. The present data indicate that ASICs emerged after the split between bilaterians (organisms with two-fold symmetry) and Cnidaria (jellyfish, anemones, corals, etc.), approximately 680 million years ago. This evolutionary and functional analysis of ASIC channels across bilaterian lineages provides relevant information about the evolution of nervous and sensory systems.

---

## [Decision Letter]

**Decision letter after peer review:**

Thank you for submitting your article "Peripheral and central employment of acid-sensing ion channels during early bilaterian evolution" for consideration by *eLife*. Your article has been reviewed by 2 peer reviewers, and the evaluation has been overseen by a Reviewing Editor and Richard Aldrich as the Senior Editor. The reviewers have opted to remain anonymous.

The reviewers have discussed their reviews with one another, and the Reviewing Editor has drafted this to help you prepare a revised submission. While we identified 8 different points that should be addressed, most of them only require clarifications, additional explanations, or further quantitative analyses of existing data. Overall, we think that this will critically improve the revised version and when addressed make it suitable for publication in *eLife*.

Essential revisions:

1. Figure 6 summarizes a lot of information. It would be much clearer if the authors included a legend in the figure for what A means, what B means and what an A with a cross through it means, what the dashed box means, etc. Also, those letters should be much bigger. And how might the figure look if the As and Bs were persistent? So bilaterian has A and B, Protostomia would just have B, Deuterstomes would have A and B, etc. That way the reader doesn't have to backtrack to what might be in that group.

Consider removing the three rightmost columns, the placozoa, porifera, and ctenophore. They have no ASICs or relevant information. They are just distractors. The cnidaria column should stay since it's the last ancestor but I suggest removing the rest.

Maybe put an estimate of MYA along the y-axis at each branch point? Although this might make the figure too busy.

2. Are there discrete regions of the channel that change between deuterostomes or xenacoelomorpha? Or regions that are completely preserved? Is this acidic pocket, palm domain, or other areas conserved? The authors have a wonderful database here which could potentially be very insightful but they barely discuss the conservation of the channel domains at all.

3. There are a few ways to functional data could be expanded upon. A table summarizing the pH50s for activation, SSD, max current, and permeability should be included. Also, is it possible the reported differences arise due to differential Ca sensitivity not apparent affinity for protons or ability for the channel to gate? And the data in Figure 3B inset shows a fairly large non-desensitizing component that seems to get bigger with more acidic pHs. Does this sustained current show a similar permeability as the peak current?

4. How confident are the authors about expression specificity? Can they consider an ASIC-sense control probe to test the specificity of ASICs in the various cells? Conversely, can the identity of the peripheral and central cells in the different organisms be validated with other cell markers? We acknowledge that this has been done, for example, in I. pulchra and T. transversa (Figure S2). But, the central vs. peripheral expression through evolution is an important result, and strengthening this data would be helpful. Please provide arguments if this is experimentally difficult.

5. Along the lines of expression, please indicate how many times in situ was performed and whether a similar expression profile was achieved across replicates.

6. What is the difference in sequence identity between the tested ASICs from different species? Since this is the first study to characterize these ion channels, more information about protein sequence identity between the ASIC channels from the different species tested will be useful to the readers.

7. Figure 3A; I. pulchra, ASIC currents look like are outward after activation. The baseline current before the stimulus is much lower than the current during stimulus application. The authors should explain this profile. Longer traces and more signal representation (before and after stimulus application) should be provided.

8. The inactivation kinetics among the different tested ASIC channels are diverse. The authors should quantify these properties and report average values across the channels from different species.

---

## [Author Response]

Essential revisions:1. Figure 6 summarizes a lot of information. It would be much clearer if the authors included a legend in the figure for what A means, what B means and what an A with a cross through it means, what the dashed box means, etc. Also, those letters should be much bigger. And how might the figure look if the As and Bs were persistent? So bilaterian has A and B, Protostomia would just have B, Deuterstomes would have A and B, etc. That way the reader doesn't have to backtrack to what might be in that group.Consider removing the three rightmost columns, the placozoa, porifera, and ctenophore. They have no ASICs or relevant information. They are just distractors. The cnidaria column should stay since it's the last ancestor but I suggest removing the rest.Maybe put an estimate of MYA along the y-axis at each branch point? Although this might make the figure too busy.

Figure 6 has been updated to include these suggestions. We didn’t add a MYA scale, because we agree the figure would be too busy.

2. Are there discrete regions of the channel that change between deuterostomes or xenacoelomorpha? Or regions that are completely preserved? Is this acidic pocket, palm domain, or other areas conserved? The authors have a wonderful database here which could potentially be very insightful but they barely discuss the conservation of the channel domains at all.

An amino acid sequence alignment comparing different regions of the channel in various homologues was added as a new supplementary figure (Figure 2—figure supplement 2) and the following description was added to the main text (upper page 4 of revised, tracked changes manuscript).

“Two clusters of residues in the extracellular domain, one in the acidic pocket and one in the palm/wrist domain, contain protonatable residues that are required for channel activation (Jasti et al., 2007; Paukert et al., 2008; Rook et al., 2021; Vullo et al., 2017). Residues H73, D78, E411 and E416 (rat ASIC1a numbering) from the palm/wrist domain and K211 located in an intersubunit palm-thumb interaction region are fairly conserved throughout all ASICs, however conservation of the acidic pocket was only obvious in chordate Group A ASICs, suggesting that this became important for the gating of e.g. rat ASIC1a more recently (Figure 2—figure supplement 2).”

3. There are a few ways to functional data could be expanded upon. A table summarizing the pH50s for activation, SSD, max current, and permeability should be included. Also, is it possible the reported differences arise due to differential Ca sensitivity not apparent affinity for protons or ability for the channel to gate? And the data in Figure 3B inset shows a fairly large non-desensitizing component that seems to get bigger with more acidic pHs. Does this sustained current show a similar permeability as the peak current?

Table 1 was added to summarize all the electrophysiological properties of the Bilaterian ASICs.

The point about calcium and apparent affinity is interesting/useful. We have performed additional dose-response experiments in the nominal absence of calcium (data into Table 1). These show that inhibition by calcium is common to all ASICs and thus not a reason for differences in pH50, etc. These results are described in the main text (lower-page 5 and mid-page 7 of revised manuscript).

We have analyzed the permeability of the sustained current for *Hofstenia* ASIC, showing no difference compared to the peak current. These values were added to Table 1 and are mentioned in main text, lower page 5 of revised manuscript.

4. How confident are the authors about expression specificity? Can they consider an ASIC-sense control probe to test the specificity of ASICs in the various cells? Conversely, can the identity of the peripheral and central cells in the different organisms be validated with other cell markers? We acknowledge that this has been done, for example, in I. pulchra and T. transversa (Figure S2). But, the central vs. peripheral expression through evolution is an important result, and strengthening this data would be helpful. Please provide arguments if this is experimentally difficult.

These are fair points. We have now performed additional in situ hybridization and hybridization chain reaction (similar to in situ hybridization) to verify ASIC expression patterns and to potentially assess cell types. Numerous fluorescent in situ hybridization – some using two and three markers on the same sample – results have been added to the main figures and to the figure supplements, and the main text has been updated accordingly, e.g. lower-page 4 and mid-page 6 in revised manuscript. Material and Methods also updated to describe this technique on lower-page 18 of revised manuscript.

“We think this indeed helps resolve the central/peripheral question (mostly providing more empirical confirmation of original interpretation), such as the spiralian ASICs overlapping much more with digestive system markers than with central nervous system markers (Figure 4 and Figure 4—figure supplement 1), even though we were not always able to identify cell type per se. For one of the animals, *Isodiametra pulchra* in Figure 3A and Figure 3—figure supplement 1, we were limited by sample availability, but even here, we were able to assess our last remaining samples with an additional marker.”

We have also added to the acknowledgements several previous members of the Hejnol lab, as some of our new experiments used old clones from these previous lab members (lower-page 19 of revised manuscript).

5. Along the lines of expression, please indicate how many times in situ was performed and whether a similar expression profile was achieved across replicates.

When results were consistent (most animals), in situ hybridization was performed at least twice. For ambiguous results (*S. californicum*), several more rounds were performed, and the most representative results were chosen for display in figures. We have added this information to Materials and methods, Immunohistochemistry and situ hybridization, mid-page 18 of revised manuscript.

6. What is the difference in sequence identity between the tested ASICs from different species? Since this is the first study to characterize these ion channels, more information about protein sequence identity between the ASIC channels from the different species tested will be useful to the readers.

We have calculated amino acid sequence identity and added the following text to the main text (upper-page 4 of revised manuscript). Author response table 1 is for the reviewer’s reference, we have not added it to the manuscript.

**Author response table 1. sa2table1:** 

	Rat	Ambulacraria	Acoel	Spiralia	
Rat	Overall	48-51%	26-32%	24-31%	27-35%
	TM1	48-60%	16-44%	24-36%	20-40%
	ECD	54-74%	31-36%	31-38%	31-39%
	TM2	82-71%	38-53%	47-59%	47-62%
Ambulacraria	Overall		35-56%	22-23%	27-30%
	TM1		16-52%	12-24%	12-28%
	ECD		42-63%	27-29%	31-41%
	TM2		47-76%	32-44%	35-56%
Acoel	Overall			49%	22-28%
	TM1			84%	12-28%
	ECD			55%	31-36%
	TM2			85%	38-56%
Spiralia	Overall				40-42%
	TM1				20-32%
	ECD				42-45%
	TM2				59-65%

“Newly described ASICs are 24-35% identical to rat ASIC1-3, the latter of which share 48-51% identity with each other. The second transmembrane helix is the most conserved segment (38-62%), followed by the extracellular domain (31-39%), with the first transmembrane helix being the most diverse (14-44%).”

7. Figure 3A; I. pulchra, ASIC currents look like are outward after activation. The baseline current before the stimulus is much lower than the current during stimulus application. The authors should explain this profile. Longer traces and more signal representation (before and after stimulus application) should be provided.

We are grateful this was pointed out, we revisited this by showing current recordings with less severe filtering and by performing additional experiments to confirm.

Previously we had filtered conservatively for analysis (200 Hz) which lost no significant signal, but we perhaps filtered too severely when filtering further for figures to reduce filesize (previously 1 Hz for all ASICs, now 1 or 10 Hz depending on the ASIC – now described on mid-page 19, Electrophysiological recordings and data analysis).

Figure 3A is now updated to show a better recording and more signal (less filtering), which shows the correct, larger amplitude of the transient current relative to the sustained current and also better resolves the direction of the sustained current. We have also analysed the relative ion permeability of the transient and sustained current and find they are similar (data into Table 1). We have updated the relevant main text to read as follows (mid-page 5 of revised manuscript.)

At oocytes expressing I. pulchra ASIC, drops to pH 5.6 and lower rapidly activated a transient current (Figure 3A, mid-right). However, compared to other ASICs, responses to increased proton concentrations were relatively inconsistent at I. pulchra ASIC: although transient currents were never activated by pH higher than 5.9, the concentration dependence of the transient current between pH 5.6 and 4.0 was inconsistent and at lower pH was usually followed by a sustained current ~25% the amplitude of the transient current (Table 1).

8. The inactivation kinetics among the different tested ASIC channels are diverse. The authors should quantify these properties and report average values across the channels from different species.

We have now analyzed the decrease in current amplitude that occurs during increased proton concentration (desensitization) by calculating the time taken for the peak current amplitude to decrease to 50% during prolonged proton application (T_50%_). The results have been added to Table 1 and described in the main text, and indeed these kinetics differ from ASIC to ASIC, particularly for *Lingula anatina* (a brachiopod spiralian) ASIC, which we now describe as follows (upper-page 7 of revised manuscript).

*L. anatina* ASIC was sensitive to relatively low proton concentrations, with large currents in response to pH 7.4 and lower that enter desensitization slower than most ASICs (Figure 4A, pH_50_ = 7.3 ± 0.2, time to 50% current amplitude (T_50%_) = 8.96±0.42 seconds, Table 1) and showed typical Na^+^/K^+^ selectivity (Figure 4A, P_Na+_/P_K_^+^ = 9.0 ± 1.7). *L. anatina* ASIC has a cysteine residue in contrast to an asparagine residue at position 414 (rat ASIC1a numbering) of vertebrate ASICs (brown in Figure 2—figure supplement 2A). This cysteine residue might contribute to the slow desensitization of *L. anatina* ASIC, as the N414C mutation in human ASIC1a (rat ASIC1a numbering) slows desensitization slightly and its chemical modification slows desensitization greatly (Roy et al., 2013).